# A Theory of Data Acquisition and Pricing at Scale

**Andrew Ilyas** [1]  **Amin Saberi** [2]  **Grigoris Velegkas** [3]

## Abstract

Data plays an invaluable role in large-scale ML training pipelines. Multiple factors, including the need to incentivize the creation of high-quality data and efforts to compensate creative data work, have led to increased interest in data *pricing*. Data pricing mechanisms seek to establish a market where data providers are compensated based (in part) on the value of their data to the data buyer, e.g., frontier AI labs. However, assessing the exact value that each provider's data adds to the data buyer's objective requires repeated re-training, which is infeasible in practice. Our work studies *data pricing under compute constraints*. In our setting, data buyers cannot make data acquisition decisions optimally due to limited compute. Inspired by existing practice in the field of data selection, we propose a model for this problem called "pricing with an attribution oracle," and provide a theoretical analysis of compute-efficient acquisition and pricing.

## 1. Introduction

Training data composition is a crucial component of the large-scale machine learning pipeline. While the open web provides a large corpus of freely accessible training data, many high-quality datasets are proprietary, protected by copyright, or costly to collect and label. Thus, the providers of such datasets may not want to share them without adequate compensation. This motivates the following *data pricing* problem.

Consider $k$ strategic *data sources* (sellers); each seller $i$ can generate samples from a distribution $D_i$ at a (private) cost of $c_i$ per sample. Fix a data budget $B > 0$. A *data buyer* who

___________

[1]Carnegie Mellon University, Pittsburgh, Pennsylvania, USA [2]Stanford University, Stanford, California, USA [3]Google Research, Mountain View, California, USA. Correspondence to: Andrew Ilyas <andrewi@andrew.cmu.edu>, Amin Saberi <saberi@stanford.edu>, Grigoris Velegkas <gvelegkas@google.com>.

*Proceedings of the 43rd International Conference on Machine Learning*, Seoul, South Korea. PMLR 306, 2026. Copyright 2026 by the author(s).

aims to train a machine learning model sees the distributions $D_i$ and *reported* costs $\hat{c}_i$ (which, crucially, may not coincide with the true per-sample costs $c_i$). With this information, the buyer decides on the amount of data $x_i$ it will purchase from each seller $i \in [k]$ as well as the per-sample price $p_i \geq \hat{c}_i$, with the goal of maximizing the expected performance of their machine learning model subject to a budget constraint. The *data pricing mechanism*—the mapping from $\{(D_i, \hat{c}_i)\}$ to $\{x_i, p_i\}$—is known to the providers a priori, whose goal is to maximize their own profit, measured as $x_i \cdot (p_i - c_i)$. Such a mechanism is called *incentive-compatible* if the data providers maximize their utility by reporting their *true* costs.

The buyer wishes to design a mechanism that allows them to train a high-quality machine learning model without imposing undue costs on the data sellers. A prerequisite to designing such a mechanism is being able to solve the corresponding *allocation* problem: suppose the buyer saw both the distributions $D_i$ and were *given* (posted) prices $p_i$; how should they purchase data? This problem turns out to be a well-studied one. For example, the field of optimal experimental design (Hashimoto, 2021) studies the relationship between covariate design and expected error in statistics. Most recently, this problem can be framed as an instance of data selection-based learning (Kolossov et al., 2023). However, the vast majority of prior works does not account for the *computational cost* that is associated with solving the data selection problem, even in the absence of incentives.

Our contribution is to make this computational bottleneck part of the mechanism-design problem, rather than a downstream implementation detail. Prior work typically studies either compute-aware data selection without strategic sellers, or strategic data acquisition assuming that the buyer can evaluate allocations exactly. Combining the two is not immediate: once the buyer uses an approximate or early-stopped allocation rule, monotonicity can fail and truthful payments may cease to exist. We therefore study the joint compute–truthfulness–utility tradeoff induced by oracle-based allocation.

### 1.1. Informal model

In our work, we propose a model of data selection and pricing under compute constraints. The main motivation behind our model is that

(a) in theory, one can often characterize the optimal data selection (e.g., using a closed-form expression, or implicitly via fixed-point conditions); *but*,

(b) computational constraints often prevent data buyers from making theoretically optimal data selections; *so*,

(c) practitioners make use of coarse *data filtering* heuristics to select data.

Motivated by this observation, our model restricts the buyer to acquisition procedures that are driven by explicit *data filtering/attribution heuristics*. On top of this computational constraint, the buyer must also set payments so that the overall protocol is *truthful*.

**Data selection heuristics as loss predictors.** To design a formal model inspired by the aforementioned practical considerations, we first need to decide on how to represent the space of possible data filtering heuristics. In this work, we consider a broad class of heuristics that we abstract as *data attribution oracles*. Informally, a data attribution oracle takes as input a candidate data mixture (or acquisition vector) $\vec{n}$ and returns an estimate of the buyer's learning objective (and, in our formal model, an estimate of its *gradient* with respect to the mixture). This idea captures a wide variety of data filtering heuristics, including:

(a) training proxy models (e.g., at smaller scale or on less data) across many candidate mixtures (Ye et al., 2024; Liu et al., 2024; Belenki et al., 2025) to get a rough estimate of data quality;

(b) iteratively pruning low-quality data using duplicate detection (Lee et al., 2021; Penedo et al., 2023) or example-filtering classifiers (Raffel et al., 2020);

(c) measuring similarity with benchmarks (Moore & Lewis, 2010) or known high-quality data sources (Wenzek et al., 2019; Schuhmann et al., 2022) to assess the quality of each data provider;

(d) leveraging approximations of the influence function (Hampel, 1974; Koh & Liang, 2017) or Shapley value (Shapley et al., 1953; Ghorbani & Zou, 2019) to estimate the marginal effect of adding or removing data from each source.

The main differences between the heuristics are in *computational cost* and *fidelity*. For example, filtering obviously low-quality data is inexpensive yet coarse, while fitting a scaling law may give more accurate predictions of model behavior but can also be quite expensive.

**Modeling the cost-of-compute.** Another element we need to consider is the relationship between compute costs and data costs, which is highly uncertain. On one hand, as compute continues to get cheaper there is some evidence that the cost of running these optimization heuristics is getting cheaper. On the other hand, if synthetic data proves to be effective or algorithms become significantly more data-

efficient, then this ratio might stay constant. To account for this uncertainty, our model is parameterized by a variable $\lambda$ that we call the *cost-of-compute*: in our case, $\lambda$ corresponds to the "oracle compute units" and monetary units.

Apart from data selection, we also need to account for the *incentives* of the participating sellers. In single-parameter procurement settings, Myerson's seminal result (Myerson, 1981) implies that an allocation rule is implementable as a truthful mechanism if and only if it is *monotone* (in procurement form): when seller $i$ reports a *lower* cost (holding other reports fixed), the buyer's purchased quantity from $i$ is (weakly) *higher*. Moreover, the corresponding truthful payments are given by an integral identity (Myerson's payment formula). In our model, computing these payments can introduce nontrivial computational overhead, which we explicitly include as part of the total compute cost.

**Informal model.** Having discussed the previous considerations, we can now give an informal description of our model. We have $k$ data sources, each having a data distribution $D_i$ with corresponding per-sample cost $c_i$, and a single data buyer who employs a learning algorithm $\mathcal{A}$ and has a *utility function $U$*. The interaction proceeds as follows:

1. First, the buyer commits to a *data acquisition protocol* that maps reported costs $(\hat{c}_i)_{i \in [k]}$ (and the budget $B$) to allocation vectors and payments. Importantly, the description of the protocol can be done *implicitly*, i.e., without an exhaustive communication of these functions. Moreover, the buyer might have access to public information about the sellers; for instance, it might know the distribution of the data they produce.

2. The sellers each report a per-sample cost $(\hat{c}_i)_{i \in [k]}$.

3. After observing the costs, the buyer executes the data acquisition protocol. In our setting, this involves running an optimization algorithm using a sequence of *loss prediction oracle calls* $[O_1, \ldots, O_T]$ and an *update function* $s : \mathbb{R}^k \times \mathbb{R}^k \to \mathbb{R}^k$. In more detail, the data acquisition algorithm looks as follows:

   (a) Initialize a vector $\pi_0 \in \mathbb{R}^k$ representing an initial guess at how to acquire data.

   (b) For $t \in [T]$:
       i. Compute $u_t = U(\pi_t)$ and $g_t = \mathcal{O}_t(\pi_t)$
       ii. Update $\pi_{t+1} = s(\pi_t, g_t)$

   (c) Output $\pi_T$ as well as a vector of per-sample *payments* $\mathbf{p} = \mathcal{P}(\hat{c}, \{(\pi_t, u_t, g_t)\}_{t=1}^T)$.

4. The buyer receives a final utility of the form (loss of model) + (compute and data cost).

At this level of generality the interaction protocol looks very abstract, thus it is useful to instantiate its components. In

this setting, after observing the reported per-sample cost of the data, the buyer starts with some arbitrary data mixture $\pi_0$. Then, it applies a first-order optimization algorithm (e.g., gradient descent) on the space of data mixtures. Since computing the gradient of the loss function with respect to the data mixture weights is often as expensive as retraining from scratch the buyer utilizes the loss prediction oracles to get an *estimate* of the gradient; above, we denote by $O_t$ the oracle employed at the $t$-th, $g_t$ the estimated gradient, and $s(\pi t - 1, g_t)$ the updated mixture after taking one step of the optimization algorithm.

A crucial building block in designing a truthful mechanism is to first design an allocation function without considering the payments at all, i.e., by pretending that the sellers report truthfully to the mechanism. Then, we will explain how to design appropriate payment rules.

## 1.2. Informal Results

By instantiating our model in the context of *linear regression* we are able to obtain several *qualitative* insights about this compute-aware setting and develop technical tools that could be useful in other contexts. In this setting, the optimal data-mixture has a closed form solution, hence one might think that it is rather straightforward to solve the problem optimally and derive appropriate payments using Myerson's result (Myerson, 1981). Rather surprisingly, it turns out that even in this fundamental ML setting there is a rich theoretical landscape when one takes into account the cost of compute. At this point the astute reader might wonder, how the cost of compute is defined when the problem has a closed form solution? Crucially, the closed-form solution involves inverting data-dependent matrices, whose dimension depends on the ambient dimensionality of the data, as well as the size of the training set. This makes such operations extremely costly for large-scale ML applications. Below, we summarize the main conceptual takeaways from our results.

**Gradient oracle for linear regression.** An important building block for our results is an oracle that estimates the gradient of the loss function with respect to the weight of the data mixture at a desired granularity $\epsilon$. We design such an oracle for the task of linear regression and prove its cost-compute guarantees. Importantly we show that if we instantiate this oracle with vanishing error-rates and we ran gradient descent with the approximated gradients, the solution converges to the optimal data mixture. We believe this could be of interest outside the context of our work.

**Selection is compute-sensitive.** Next, we show that the optimal data selection policy depends heavily on the cost of compute and the data heterogeneity. For this set of results, we assume that we can obtain exact gradients at a given (fixed) cost. As a first step, we show that it is prohibitive to even perform *one* step of data mixture optimization in a certain regime of the compute cost. In its full generality, our result shows the optimal number of steps to run the gradient descent algorithm for a given compute budget, when we need to commit to the number of steps *statically*, i.e., at the beginning of the optimization process. Lastly, we show how to choose the optimal number of steps adaptively, based on the trajectory of the improvements of the loss function and the shape of the dataset.

**Optimal compute scheduling.** Subsequently, we return to the setting where the buyer has access to costly data estimation oracles that give some $\epsilon$-approximation of the gradient at cost $f(\epsilon)$. Naturally, given such oracles and a compute budget, a natural question is how should one distribute this budget during the optimization process? We derive such an optimal scheduling as a function of the cost of the oracles and the compute budget. As a corollary, we see that it is more efficient to use coarse estimations of the gradient at the earlier stages of the optimization process and finer ones later on.

**Performance-truthfulness tradeoff.** Having obtained insights about the behavior of the data selection rule in this compute-constraint setting, we turn to the problem of designing *mechanisms*. A first important observation is that compute constraints introduce subtle, but potentially catastrophic, *incentive issues*. While in the setting of linear regression it is not hard to show that solving the problem *optimally* gives rise to monotone data selection rules, in the presence of compute constraints obtaining such an optimal solution is prohibitive. Unfortunately, suboptimal solutions can lead to *non-monotone* allocation rules; this is problematic because such allocation rules do not admit *any* payments that induce truth-telling. This is pathological both for the buyer and the sellers: the buyer cannot predict what sellers will bid as a function of their true cost, and some seller with cost $c$ has to make educated guesses about its competition in order to bid optimally. Fortunately, we are able to show how to restore (approximate) monotonicity by adding a *strongly convex* component to the objective function.

**Zero-overhead procurement auctions.** As we alluded to before, when the data selection rule is (approximately) monotone, one can use Myerson's formula to come up with appropriate payments to incentive truthtelling. Unfortunately, a direct implementation of Myerson's payment formula necessitates re-running the data selection rule $k \cdot 1/\epsilon$ times, where $k$ is the number of data providers and $\epsilon$ is the desired accuracy of the payment calculation. In modern applications, this overhead is prohibitively expensive, as every additional execution might cost hundreds of millions of dollars. To circumvent this barrier, we build upon results from the literature of item auctions (Babaioff et al., 2015) to show how to compute both the data selection and the payments with a *single* call to the data selection rule! This

might be of interest outside the scope of our work.

## 1.3. Overview of Techniques

At a high level, our main results leverage techniques spanning statistics, optimization, numerical linear algebra, and mechanism design. On the statistical front, we propose a continuous parameterization of the data acquisition problem, and use tail bounds together with Matrix Bernstein-style inequalities (Kereta & Klock, 2021) to relate this objective (and its gradient) to its population counterpart. Using tools from numerical linear algebra (namely, stochastic trace estimation (Meyer et al., 2021) and conjugate gradient methods), we show how to efficiently estimate the gradient of this population objective. Together, these techniques allow us to describe the behavior of the buyer as a perturbation of a smooth convex optimization problem with a noisy gradient oracle. This point of view yields a number of our main results, which involve analyzing the properties (e.g., convergence rate, monotonicity, etc.) of both converged and non-converged solutions to such optimization problems. Finally, when turning our attention to the problem of designing truthful mechanisms, we will extend existing results from the item auction literature (Babaioff et al., 2015) to our setting. These results show how to convert, in a black-box way, an approximately monotone allocation rule to an approximately truthful mechanism without requiring any additional calls to the allocation rule.

## 1.4. Related work

**Related areas in statistics.** Classical statistical literature studies optimal experimental design, active learning, and subsampling for estimation and prediction under budget constraints. Examples include A-/D-optimal design, leverage-score and influence-based sampling, and importance sampling for risk minimization; see surveys and developments in optimal design and subsampling for large data settings (e.g., Pukelsheim, 2006; Kolossov et al., 2023). These works typically assume a known generative model or tractable risk proxy and do not model strategic providers or the *compute* required to evaluate utility at scale.

**Empirical data selection in ML.** Recent practice-driven work proposes heuristics for building high-utility mixtures without repeatedly training at scale, including (i) small-scale or proxy training to fit scaling laws and extrapolate utility (Kaplan et al., 2020; Hoffmann et al., 2022; Ye et al., 2024), (ii) deduplication and quality filtering (Lee et al., 2021; Penedo et al., 2023; Raffel et al., 2020), and (iii) attribution via influence functions or data Shapley approximations (Hampel, 1974; Koh & Liang, 2017; Ghorbani & Zou, 2019). While effective empirically, theoretical guarantees often hold under stylized surrogate objectives or assume inexpensive evaluation. Our data attribution oracle abstraction is designed to capture these procedures uniformly while making the compute–fidelity tradeoff explicit.

**Data valuation and attribution.** A growing line of work studies how to apportion credit or value to training examples or sources. Influence-function approaches estimate the effect of upweighting or removing a point on model parameters or risk (Hampel, 1974; Koh & Liang, 2017); Shapley-based methods target cooperative-game values for examples or groups (Shapley et al., 1953; Ghorbani & Zou, 2019; Jia et al., 2019). Several recent methods accelerate these computations, but most still presuppose repeated access to gradients or retraining loops that become prohibitive at frontier scales. In our framework, such methods correspond to higher-fidelity, higher-cost data attribution oracles.

**Data pricing and markets.** There is rich economic theory on selling information and designing data markets (Bergemann & Bonatti, 2019; Kamenica & Gentzkow, 2011; Cai & Velegkas, 2020). Recent papers study how to price data for learning objectives, budget-feasible procurement, and contracts for information that depends on hard-to-observe quality (e.g., Agarwal et al., 2019; Cong et al., 2022; Balkanski et al., 2022). A common idealization is that the buyer's value is either known or cheaply verifiable; our contribution is to make the compute needed to *evaluate* value a first-class resource via data attribution oracles and design mechanisms that are faithful to incentive goals under this constraint.

## 2. Formal Model

We now formalize the model described informally in Section 1.1. The following sections instantiate this model (most notably for linear regression) and present our main results.

**Definitions and setup.** We consider a setting with $k$ data sources (sellers), each corresponding to a distribution $\mathcal{D}_i$ over covariate-label pairs $(\boldsymbol{x}, y)$, where $\boldsymbol{x} \in \mathbb{R}^d$ and $y \in \mathcal{Y}$. We write $[k] \triangleq \{1, 2, \ldots, k\}$. Recall that $\mathcal{D}_i, i \in [k]$, is known to the buyer. This captures settings where the buyer has a public view, metadata, summary statistics, or an audit sample of the source, but must pay for commercial use, labels, or curation. Each source $i$ has a (private) per-sample cost $c_i \in \mathbb{R}_{>0}$. The buyer has a capital (data) budget $B > 0$.

The buyer has a (potentially random) learning algorithm $\mathcal{A} : (\mathcal{X} \times \mathcal{Y})^* \to \Theta$ and evaluates the trained model via a loss function $L : \Theta \to \mathbb{R}$. For an allocation vector $\vec{n} \in \mathbb{Z}_+^k$, let $S(\vec{n}) \in (\mathcal{X} \times \mathcal{Y})^{\|\vec{n}\|_1}$ be the random dataset obtained by drawing $n_i$ i.i.d. samples from each source $\mathcal{D}_i$ and pooling.

**Poissonized acquisition.** Primarily for mathematical convenience, we parameterize the buyer's acquisition decision directly by a weight vector $\pi \in \Delta^k$, where $\Delta^k = \left\{ \pi \in \mathbb{R}_{\geq 0}^k : \sum_{i=1}^k \pi_i = 1 \right\}$. Here, $\pi_i$ represents the fraction of the buyer's budget allocated to source $i$. That is,

for fixed per-sample prices $\mathbf{p} = (p_1, \ldots, p_k)$, the realized sample counts are drawn as $n_i \sim \text{Poisson}\left(\frac{B\pi_i}{p_i}\right)$. Equivalently, $\mathbb{E}[n_i] = B\pi_i/p_i$ and $\mathbb{E}[p_i n_i] = B\pi_i$, so $\sum_i \pi_i = 1$ enforces $\mathbb{E}[\sum_i p_i n_i] = B$. The buyer trains on $S(\vec{n})$, and evaluates the (scaled) expected loss function $f : \Delta^k \to \mathbb{R}$ $f(\pi) = B \cdot \mathbb{E}[L(\mathcal{A}(S(\vec{n})))]$, where the expectation is over the data draws, the sampling of $\vec{n}$, and any randomness in the learning algorithm $\mathcal{A}$.

### 2.1. The data acquisition problem

We will first study the *data acquisition* problem, where the buyer knows the true per-sample costs. For notational clarity, we treat $\mathbf{p} = (p_1, \ldots, p_k)$ as exogenous posted per-sample prices; as a full-information benchmark we set $\mathbf{p} = \mathbf{c} = (c_1, \ldots, c_k)$.

**Budget.** Given posted prices $p_i = c_i$ and budget $B$, the buyer chooses $\pi \in \Delta^k$ and the expected dollars spent on source $i$ is $B\pi_i$, and the realized sample counts are $n_i \sim \text{Poisson}(B\pi_i/p_i)$. Thus, $\sum_i \pi_i = 1$ enforces that the buyer's expected spend on data is $B$.

**Gradient access and compute cost.** As discussed in Section 1.1, even a single run of the learning algorithm $\mathcal{A}$ may be expensive to compute, and so in practice the buyer will not be able to perfectly minimize the expected loss function $f(\pi)$. Instead, practical acquisition algorithms are driven by *data filtering heuristics*, which give estimates of the marginal value of data from each source. We model such marginal information via a single *data attribution oracle* parametrized by an accuracy level $\epsilon$: querying the oracle at accuracy $\epsilon$ incurs a known compute cost $\gamma(\epsilon)$.

**Definition 1** (Gradient Approximation Oracle)**.** *Let* $f :$ $\Delta^k \to \mathbb{R}$ *be the buyer's loss function (defined above). A* ***data attribution oracle*** $O$ *is a map* $O : \Delta^k \times \mathbb{R}_{>0} \to \mathbb{R}^k$ *where for any acquisition vector* $\pi \in \Delta^k$ *and desired accuracy* $\epsilon \in \mathbb{R}_{>0}$, *the oracle returns a vector* $\tau = O(\pi, \epsilon)$ *satisfying* $\|\tau - \nabla f(\pi)\|_2 \leq \epsilon$. *Accessing this oracle incurs a compute cost given by a* ***cost function*** $C : \mathbb{R}_{>0} \to \mathbb{R}_{\geq 0}$, *which is publicly known and* ***non-increasing*** *in* $\epsilon$.

Notice that this definition does not require that the oracle is unbiased. Moreover, we remark that we also allow for randomized oracles that satisfy analogous guarantees with high probability. We assume that the buyer runs projected gradient descent using successive noisy gradient estimates returned by the oracle at accuracies $\epsilon_1, \epsilon_2, \ldots$ and may stop at any time $t$. If the algorithm stops at time $t$ and outputs $\pi_t$, then the total compute spend is $\sum_{s=1}^{t} \gamma(\epsilon_s)$ and we evaluate the buyer's compute-aware objective by $f(\pi_t) + \lambda \sum_{s=1}^{t} \gamma(\epsilon_s)$, where $\lambda > 0$ is a cost-of-compute parameter. Recall that $\lambda$ is a parameter that translates loss units to compute / monetary units. Ignoring compute cost, the buyer's acquisition problem is to find

$\arg\min_{\pi \in \Delta^k} f(\pi)$. The central computational question is how to approximately solve it while accounting for the compute cost of obtaining gradient information (i.e., how to choose the accuracy schedule $\{\epsilon_t\}$, and when to stop).

### 2.2. The mechanism design problem

Having treated the case where per-sample costs (prices) are known, we now consider the more realistic setting where costs are private to the sellers. Seller $i$ has a private per-sample cost $c_i$ and reports $\hat{c}_i$ to the buyer. The buyer commits to a protocol that maps reports $\hat{c}$ to (i) per-sample prices (or, more generally, payments) and (ii) an acquisition procedure (including an oracle accuracy schedule and a stopping rule) that determines how much data to procure from each seller. Similarly as before, the objective of the buyer is to minimize $\mathbb{E}[f(\pi(\hat{c}))] + \lambda \sum_{t=1}^{T} \gamma(\epsilon_t)$, subject to the constraint $\sum_{i=1}^{k} \pi_i = 1$. The samples $n_i$ from source $i \in [k]$ are drawn as $n_i \sim \text{Poisson}\left(\frac{B\pi_i}{c_i}\right)$. This implies the budget constraint holds (in expectation over the random draws according to the mixture $\pi$), i.e., $\mathbb{E}[\sum_{i=1}^{k} c_i \cdot n_i] = B$. Notice that the budget constraint is on the total cost that the data sources incur to label data and not on the payments. This is because in modern applications, even if buyers are willing to spend an enormous amount of money on payments there are certain bottlenecks on the data generators side that cannot be circumvented. Each seller $i$ chooses its report to maximize its expected profit, $\mathbb{E}[p_i(\hat{c}) n_i] - c_i \mathbb{E}[n_i]$. We study whether the buyer can design protocols that are truthful (DSIC) and individually rational while achieving good learning performance; for formal definitions (including randomized allocations) see Appendix A.

## 3. Main Results and Proof Overviews

To motivate the framework and assumptions above, we now consider a concrete instantiation of the setup, where the data buyer is interested in learning a linear model $\theta \in \mathbb{R}^d$ from unbiased data sources. We start in the simplest such setting, and then explore extensions to the model. Due to space constraints, most of the formal results and proofs appear in the corresponding sections of the appendix.

### 3.1. Setup: Well-Specified OLS

In this setting, $\Theta = \mathbb{R}^d$ and $\mathcal{A}$ is the ordinary least squares (OLS) estimator, which given a dataset $S$ returns a parameter vector $\theta$ that minimizes the squared error loss. Formally, $\mathcal{A}(S) = \arg\min_{\theta \in \Theta} \sum_{(x_i, y_i) \in S} (y_i - \theta^\top x_i)^2$

**Data sources.** Each data source $i$ is a product distribution of covariates sampled from a per-source distribution $x_{ij} \sim \mathcal{D}_i$, and labels $y_{ij} = \theta_\star^\top x_{ij} + \epsilon_{ij}$ where $\theta_\star \in \Theta$ is the true parameter and $\epsilon_{ij} \sim \mathcal{N}(0, \sigma_i^2)$ are independent noise terms.

The buyer's utility from a parameter $\theta$ is then given by the squared loss on a held-out test set with identity covariance, meaning that $u(\theta) = \|\theta - \theta_*\|_2^2$.

**Common notation and assumptions.** We assume that the random variable $X_i \sim \mathcal{D}_i$ is mean-zero and $\sigma_x$-subgaussian: for all $u \in \mathbb{R}^d$, $\mathbb{E}\left[e^{u^\top X_i}\right] \leq \exp\left(\frac{\sigma_x^2 \|u\|_2^2}{2}\right)$. We write the second moment matrix of source $i$ as $\Sigma_i := \mathbb{E}[X_i X_i^\top] \in \mathbb{S}_+^d$ and assume uniform spectral bounds $mI \preceq \Sigma_i \preceq MI$ as well as an absolute upper bound on the costs $c_{max}$. For a fractional acquisition vector $\pi \in \Delta_k$, and prices $p \in \mathbb{R}_{\geq 0}^k$ we use $w(\pi) \in \mathbb{R}_{\geq 0}^k$ [1] to denote the vector of the expected number of samples from each source, i.e., $w_i = B\pi_i/p_i$. We define the population information matrix $S(w(\pi))$ and the empirical information matrix $\widehat{S}(\vec{n})$ as $S(\pi) := \sum_{i=1}^k w_i(\pi_i)\Sigma_i$, $\widehat{S}(\vec{n}) := X(\vec{n})^\top X(\vec{n})$, where $X(\vec{n})$ is the design matrix formed by stacking the sampled covariates. Then, the expected loss function is given by $f(\pi) = \mathbb{E}_{\vec{n} \sim \text{Poisson}(w(\pi))}[\text{tr}(\widehat{S}(\vec{n})^{-1})]$. For convenience, we will define a *surrogate* loss function $\widehat{f}(\pi)$ as $\widehat{f}(\pi) = \text{tr}(S(\pi)^{-1})$. Finally, define the (population) leverage scores $s_i(\pi) := \text{tr}(S(\pi)^{-1}\Sigma_i S(\pi)^{-1})$, their centered version $v_i(\pi) := s_i(\pi) - \frac{1}{k}\sum_{j=1}^k s_j(\pi)$, and the induced direction matrix $D(\pi) := \sum_{i=1}^k v_i(\pi)\Sigma_i$.

**Computational model.** We operate under a computational model where, for a matrix $X \in \mathbb{R}^{d_1 \times d_2}$, the cost of matrix-vector products $u^\top X$ and $X^\top v$ for $u \in \mathbb{R}^{d_1}$ and $v \in \mathbb{R}^{d_2}$ is $d_1 \times d_2$. We assume that the second moment matrices $\Sigma_i$ are known by the buyer (since the price of pre-computing them does not scale with the number of oracle queries).

### 3.2. Data attribution oracles

Recall that a loss estimation oracle is a function that takes as input an acquisition vector $\pi$ and returns a vector $\tau$ such that $\|\tau - \nabla f(\pi)\|_2 \leq \epsilon$. at an oracle cost $\gamma(\epsilon)$. We can derive that the true gradient of the expected loss function $f(\pi)$ at coordinate $i$ is given by

$$\nabla_i f(\pi) = -\mathbb{E}_{\substack{\vec{n} \sim \text{Poisson}(\pi) \\ X \mid \vec{n}, \, x \sim \mathcal{D}_i}}\left[\frac{\|(X^\top X)^{-1}x\|_2^2}{1 + x^\top(X^\top X)^{-1}x}\right]. \quad (1)$$

It turns out that when the budget $B$ is large enough, we can construct a family of $\epsilon$-approximate oracles by leveraging tools from numerical linear algebra. In particular, as $B \to \infty$, the behavior of (1) will be dominated by the numerator. Standard concentration bounds for the sample covariance matrix imply that this numerator will converge to $\mathbb{E}[\|S(\pi)^{-1}x\|_2^2]$, which we can rewrite as $\mathbb{E}[\text{tr}(\Sigma_i \cdot S(\pi)^{-2})]$. Naively, we can compute this quantity at a cost of $O(d^3)$ by inverting and squaring the precomputed second moment matrices. However, we can in-

stead use a combination of Hutchinson's estimator and the conjugate gradient method to compute this quantity to precision $\epsilon$ at a cost of $O(kd^2\sqrt{\kappa}/\epsilon)$ where $\kappa = M/m$ is the condition number of the second moment matrices.

---

**Algorithm 1** Gradient Computation for Linear Regression

**Require:** Acquisition vector $\pi$, data budget $B$, precomputed moments $\{\Sigma_i^{1/2}\}$, conjugate gradient method CG : $A, v \mapsto A^{-1}v$, Hutch++ trace estimator HUTCH++ : $[u \to Au], \epsilon \mapsto \text{tr}(A) \pm \epsilon$
**Ensure:** Approximate gradient vector $\nabla f(\pi)$
1: Compute $S := S(\pi) = \sum_{i=1}^k \pi_i \Sigma_i$
2: **for** each source $i = 1$ to $k$ **do**
3:     Define $A_i(v) := \Sigma_i^{1/2}\text{CG}(S, \text{CG}(S, \Sigma_i^{1/2}v))$
4:     // This implements matrix-vector product $v \mapsto \Sigma_i^{1/2}S^{-2}\Sigma_i^{1/2}v$ at cost $O(d^2\sqrt{\kappa})$
5:     Compute $g_i := -\text{HUTCH++}(A_i, \epsilon/B)$
6:     // Returns $-\text{tr}(A_i) \pm \epsilon$ at cost $O(B \cdot \text{matvec cost}/\epsilon)$
7: **end for**
8: **return** $\{B \cdot g_i\}_{i=1}^k$ // Total cost is $O(kd^2\sqrt{\kappa}/\epsilon)$

---

The following shows that the oracle from Algorithm 1 is a $\epsilon$-approximate oracle for the expected loss function $f(\pi)$.

**Proposition 1.** *Fix $\pi \in \mathbb{R}_{\geq 0}^k$ and let $\widehat{g}(\pi, \epsilon) \in \mathbb{R}^k$ be the output of Algorithm 1, so that $\left|\widehat{g}_i(\pi, \epsilon) + B \cdot \text{tr}(\Sigma_i S^{-2})\right| \leq \epsilon$ for each $i \in [k]$. Assume the subgaussian and spectral conditions in Section 3.1 and fix $\delta \in (0, 1/2)$. If $B \geq c_{max} \cdot \max\left\{12\log\frac{4}{\delta}, 2\sigma_x^4\left(d + \log\frac{4}{\delta}\right)\right\}$, then for every $i \in [k]$ the oracle output is close to the true Poissonized gradient (1): $\left|\widehat{g}_i(\pi, \epsilon) - \nabla_i f(\pi)\right| \leq \epsilon + \frac{1024\,\sigma_x^4 d^2}{m^3 B^2} + \frac{32Md}{m^2 B}(1 + \delta)$.*

The proof of Proposition 1 can be found in Appendix B.3. The main technical tools here are the Hutchinson estimator, and then matrix concentration bounds as well as the use of the Sherman-Morrison formula to perform rank-1 updates of the inverse matrix.

### 3.3. Data acquisition for linear regression

Having explained the linear regression setup, as well as the gradient oracle we will use, we next describe our main results for *data acquisition* in linear regression, when the prices of the sellers are publicly known. Next, we will move to the more challenging setting of private costs.

**When does data selection converge?** We first show that, as long as the bias of the attribution methods decays to zero, the resulting data selection will be approximately optimal.

**Proposition 2.** *Let $f(\pi)$ be the expected loss function (see Section 3.1), and let $\pi^*$ be the minimizer of $f(\pi)$ over the budget-feasible set. Let $O$ be the oracle given by Algorithm 1 with fidelity guarantees $\{\epsilon_t\}$. Suppose the cost-of-compute $\lambda = 0$. Then, as $B \to \infty$, the average acquisition $\overline{\pi}_t$*

---

[1] When $\pi$ is clear from context we might omit it.

satisfies $f(\overline{\pi}_t) - f(\pi^*) \leq O(\frac{1}{t}\sum_{s=1}^{t}\epsilon_s)$.

The full proof of Proposition 2 can be found in Appendix B.4. Briefly, the proof proceeds as follows: we show that the oracle from Algorithm 1 is a $\epsilon$-approximate oracle for a (deterministic) $A$-optimal design objective $\widehat{f}(\pi) = \text{tr}(S(\pi)^{-1})$, and so oracle-induced gradient descent converges to an approximate minimizer of $\widehat{f}(\pi)$. We then relate $\widehat{f}(\pi)$ to the expected loss function $f(\pi)$ through a standard matrix concentration bound.

**When is data selection worth the cost?** Recall our original motivation is not to find the absolute optimal data selection, but rather the *compute-optimal* data selection. In the presence of compute constraints, the data buyer may wish to stop their optimization early to avoid paying the computational cost of more gradient updates. Indeed, it turns out when the cost of compute is high and the data sources are somewhat homogenous, the data buyer may opt to forgo data selection altogether, and use the budget to purchase more data, as captured by the following result:

**Proposition 3.** *Let $\pi_0 = \{\frac{B}{c_i k}\}_{i=1}^{k}$, step size $\eta > 0$, and $\lambda > 0$ be the cost-of-compute parameter (see Section 1.1). If the first projected step is interior (so $\pi_1 := \pi_0 + \eta v$ has nonnegative coordinates) and $\lambda \geq \eta\|v\|_2^2 - \frac{\eta^2}{M^3}\|H\|_F^2$, the compute-optimal choice is to forgo data selection.*

The full proof of Proposition 3 can be found in Appendix B.5; the main idea is to study the gradient flow dynamics, which we can relate to the one-step dynamics through a time integral. When we allow the buyer to stop adaptively, the following result shows that doing so guarantees stopping at most one step after the optimal stopping time:

**Proposition 4** (One-step-overhead adaptive stopping)**.** *Fix a convex, differentiable, $L$-smooth objective $\widehat{f} : \mathbb{R}^k \to \mathbb{R}$ and step size $\eta \in (0, 1/L]$. Assume the projection is inactive along the run, so $\pi_{t+1} = \pi_t - \eta\nabla\widehat{f}(\pi_t)$. For a compute cost $\lambda > 0$, define $J_t := \widehat{f}(\pi_t) + \lambda t$ and let $T^\star := \min\{t \geq 0 : J_t = \min_{s \geq 0} J_s\}$. Then the stopping rule $\hat{t} := \min\{t \geq 0 : J_{t+1} \geq J_t\}$ satisfies $\hat{t} = T^\star$, and identifying $\hat{t}$ requires at most one extra gradient step beyond $T^\star$ (it stops after forming $\pi_{T^\star+1}$).*

The full proof of Proposition 4 can be found in Appendix B.6; the core technique is to show that the per-step improvement $\Delta_t = \widehat{f}(\pi_t) - \widehat{f}(\pi_{t+1})$ is nonincreasing.

**Which oracles should we use?** The main degree of freedom for the buyer is their choice of the accuracy $\epsilon_t$ of the gradient oracle at each step. The next result shows that we should first use oracles with low fidelity, then use more expensive oracles with higher fidelity.

**Proposition 5** (Compute-aware back-loaded schedule)**.** *Recall the A-optimal surrogate $U_A(\pi) = \text{tr}(S(\pi)^{-1})$ over the* simplex $\Delta_k$. Assume $U_A$ is $\mu$-strongly convex and $L$-smooth on $\Delta_k$ w.r.t. $\|\cdot\|_2$, and assume additionally that $U_A$ is twice continuously differentiable on $\Delta_k$ (so that the Hessian exists everywhere and satisfies $\mu I \preceq \nabla^2 U_A(\pi) \preceq LI$). Let $\pi^\star \in \arg\min_{\pi \in \Delta_k} U_A(\pi)$. Fix a horizon $T \geq 1$, step size $\eta \in (0, 1/L]$, and a cost-of-compute parameter $\lambda > 0$. Run projected gradient descent $\pi_{t+1} = \Pi_{\Delta_k}(\pi_t - \eta\tau^{(t)})$, where the oracle returns $\tau^{(t)}$ satisfying $\|\tau^{(t)} - \nabla U_A(\pi_t)\|_2 \leq \epsilon_t$ at compute cost $\gamma(\epsilon_t)$. Then there exists an optimal schedule $\epsilon_0^\star \geq \epsilon_1^\star \geq \cdots \geq \epsilon_{T-1}^\star$. In particular, if $\gamma$ is nonincreasing, then $\gamma(\epsilon_t^\star)$ is nondecreasing in $t$, i.e., computation is back-loaded.*

The complete proof appears in Appendix C. We underline that the last-iterate error bound has the form $\|\pi_T - \pi^\star\|_2 \lesssim \rho^T\|\pi_0 - \pi^\star\|_2 + \sum_{t=0}^{T-1} w_t\epsilon_t$, $w_t = \eta\rho^{T-1-t}$, $\rho = 1 - \mu\eta$. The coefficient $w_t$ is exactly the *influence* of the oracle error at time $t$ on the final iterate: an error injected at time $t$ is propagated through $(T-1-t)$ subsequent gradient steps, and each subsequent step contracts distances to $\pi^\star$ by a factor $\rho$ due to $\mu$-strong convexity. Thus, early errors are "forgotten" geometrically (multiplied by many powers of $\rho$), whereas late errors are barely attenuated (e.g., $w_{T-1} = \eta$). When $\mu$ is larger (or $\eta$ is larger), $\rho = 1 - \mu\eta$ is smaller and forgetting is faster, making $w_t$ more skewed toward late iterations; this strengthens the incentive to back-load compute. Conversely, when the condition number $\kappa = L/\mu$ is large, $\rho \approx 1 - 1/\kappa$ is close to 1, the weights $w_t$ are nearly flat, and the benefit of back-loading is weaker. The compute penalty $\lambda \sum_t \gamma(\epsilon_t)$ is permutation-invariant in time, so to minimize the weighted error term one should assign *larger* errors (cheaper, coarser oracles) to *smaller* weights (earlier iterations), and reserve *smaller* errors (more expensive, higher-fidelity oracles) for late iterations where $w_t$ is largest. In particular, when the cost of the oracle has a nice behavior, e.g. follows a power law, we can find a closed form solution for the schedule. This appears in Corollary 1.

### 3.4. Mechanism design for linear regression

We now turn our attention to creating a *truthful mechanism* for this problem, i.e., a protocol that incentivizes the data sellers to report their true cost to the data buyer. In order to do that we will rely on a cornerstone result from game theory which gives us necessary and sufficient conditions for transforming acquisition algorithms to *mechanisms*. More precisely, Myerson's seminal work (Myerson, 1981) showed that a procurement rule can be coupled with a payment scheme to yield a *truthful* mechanism if and only if it is *monotone* in the cost of every seller, i.e., if the cost of some seller $i \in [k]$ decreases then the data buyer will purchase (weakly) more samples from seller $i$. We will also use a notion of *approximate monotonicity*, which quantifies how far from being monotone a data selection rule is. Moreover,

throughout this section we assume we have a hard compute budget $C$ instead of having it be part of the objective.

**Is the compute-agnostic acquisition policy monotone?**

Recall the convention $w_i(\pi_i) = B \cdot \pi_i/c_i$.[2] For the mechanism design results it is convenient to parametrize the functions with respect to $w$ instead of $\pi$. To that end, we denote $S(w) = \sum_{i=1}^{k} w_i \Sigma_i$, $f(w) = \text{tr}(S(w)^{-1})$. We first prove a monotonicity lemma for the *exact* (compute-unconstrained) solution. The procurement-form monotonicity condition requires that for each seller $j$, holding $c_{-j}$ fixed, the allocation $w_j$ should be weakly *non-increasing* in $c_j$.

**Theorem 1** (Monotonicity of the exact optimal procurement rule). *Fix $j \in [k]$ and consider two cost vectors $c^{(1)}, c^{(2)} \in \mathbb{R}_{>0}^k$ such that $c_i^{(1)} = c_i^{(2)}$ for all $i \neq j$ and $c_j^{(1)} \leq c_j^{(2)}$. Let $w^{(1)} \in \arg\min f(w)$ and $w^{(2)} \in \arg\min f(w)$ be (arbitrary) optimizers. Then $w_j^{(1)} \geq w_j^{(2)}$.*

The full proof appears in Appendix D.1 and it follows by analyzing properties of the optimal solutions of $f(w)$.

**Is the compute-sensitive acquisition policy monotone?**
Next, we study whether suboptimal solutions preserve monotonicity of the allocation rule. Unfortunately, this is not the case. In fact, even if the gradient oracles are not costly, if we do not run the optimization process for sufficiently many iterations. We give an explicit 2-seller example where the *optimal* solution is monotone but a *one-step* projected gradient method driven by an $\epsilon$-approximate gradient oracle is not. We give a concrete instance in Appendix D.2, that consists only of two sellers. Due to space constraints, we state the result here informally.

**Informal Theorem 1.** *There is an instance with two sellers such that if the problem is not solved to optimality, the allocation rule is not monotone.*

**Can we enforce monotonicity for compute-sensitive acquisition?**

We now formalize a sufficient condition for *approximate* monotonicity and show how adding a strongly convex component to the objective function can help meet it under a limited compute budget.

**Definition 2** ($\xi$-monotonicity (procurement, additive)). *Fix seller $j$ and fix $c_{-j}$. An allocation rule $w_j(\cdot, c_{-j})$ is $\xi$-monotone if for all $c_j < c_j' w_j(c_j', c_{-j}) \leq w_j(c_j, c_{-j}) + \xi$.*

Lemma 6 shows that if we can approximate the optimal solution with respect to the $\ell_\infty$ norm then we will get approximate monotonicity.

**Strong convexification.** Let $f_c(w)$ denote the buyer's acquisition objective for cost profile $c$. Under compute con-

---

[2]In the previous sections, $c_i = p_i$, but here they might differ.

straints, the buyer runs projected gradient descent with an approximate gradient oracle and stops after $T$ iterations, which need not reach the monotone optimum $w^\star(c)$. We consider adding a strongly convex term: $f_c^{(\alpha)}(w) := f_c(w) + \alpha R(w)$, where $R$ is strongly convex on the feasible region, and $\alpha > 0$ is a tunable strength. Recall that at iteration $t$, the oracle returns $\tau^{(t)}$ satisfying $\|\tau^{(t)} - \nabla f_c^{(\alpha)}(w^{(t)})\|_2 \leq \epsilon_t$ at compute cost $\gamma(\epsilon_t)$. Our main insight is that by adding the strongly convex component to the objective we achieve two things: first, since in every iteration $t$ we can compute the gradient of the regularizer without paying any compute cost, we can achieve *smaller* approximation error with respect to the gradient of $f_c^{(\alpha)}$ than $f_c$ at the same compute cost. Second, since we improve the condition number of our function we can achieve *faster* convergence to the optimal solution. Due to space constraints, we only state our result informally; for the formal details as well as a discussion on how to choose $\alpha$ and $R(\cdot)$ we refer to Appendix D.3.

**Informal Theorem 2.** *For every compute budget $C > 0$ and desired approximate monotonicity level $\xi$ there exists $\alpha > 0$ and regularizer $R(\cdot)$ such that the compute-constraint solution to $f_c^{(\alpha)}$ is $\xi$-approximately monotone.*

**Can we design a zero-overhead truthful mechanism?**

A remaining computational bottleneck is the *payment computation*. Even when an acquisition algorithm induces a monotone procurement rule, Myerson's payment identity (Theorem 2 in Appendix A) expresses each seller's payment as an *integral* of counterfactual allocations. In a black-box oracle-driven acquisition pipeline, evaluating such counterfactual allocations naïvely requires *re-running* the pipeline many times, multiplying the compute costs.

Our next result shows that we can compute *both* the allocation and (correct) payments using *a single call* to the acquisition algorithm. The mechanism achieves truthfulness *in expectation over the mechanism's own random seed*, and is *universally ex-post IR* (i.e., for every realization of the mechanism randomness, truthful sellers never incur negative utility). Our construction is a procurement adaptation of the implicit payment computation reduction in the seminal work of Babaioff et al. (2015). Since we need to adapt several of their results to our setting, we only state an informal version of our result here and we defer the formal derivations to Appendix D.4. At a high level, the construction works in the following randomized and *recursive* way: when seller submits cost $c_i$ with probability $1 - q_i$ we retain their report and they participate in the auction, and with probability $q_i$ we draw a different reported cost $\tilde{c}_i$ from an appropriate distribution; then we recurse. The payments are defined purely in terms of the output, in a randomized way. We can show that if the underlying data selection rule is monotone, this process leads to a truthful mechanism whose output is the

same as the one that does explicit payment computations, with high probability. Moreover, we can show this result is robust to allocation rules being $\xi$-approximately monotone.

**Informal Theorem 3.** *For $\xi$-approximately monotone allocation rule $A(\cdot)$ there exists a $\xi$-truthful (in expectation) mechanism $(A'(\cdot), p(\cdot))$ so that both $A'(\cdot), p(\cdot)$ can be computed with a single call to $A(\cdot)$, and, with high probability, on any input $c$ it holds that $A(c) = A'(c)$.*

## 4. Conclusion

This work frames data pricing under compute constraints using the *data attribution oracle*. We demonstrate that limited compute necessitates *back-loaded* compute spending and can break incentive compatibility. We resolve this pathology by modifying the objective and designing a payment scheme with no additional overhead. It would be interesting to extend this to nonconvex learners and explore the compute–truthfulness tradeoff in different markets.

## Impact Statement

This paper presents work whose goal is to advance the field of Machine Learning. There are many potential societal consequences of our work, none which we feel must be specifically highlighted here.

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

# A. Omitted Details from Section 2

This appendix formalizes the procurement (mechanism design) component introduced in Section 2.

**Agents, types, and actions.** There are $k$ sellers (data sources) indexed by $i \in [k]$. Seller $i$ can supply a divisible quantity $n_i \in \mathbb{R}_{\geq 0}$ of items (e.g., samples), and has a private per-unit cost (type) $c_i \in [\underline{c}_i, \overline{c}_i] \subset \mathbb{R}_{>0}$. Types are independent of the buyer and each other; no distributional assumptions are needed for dominant-strategy results. The buyer announces a direct mechanism $(n, p)$ mapping reported costs $\hat{c} = (\hat{c}_1, \ldots, \hat{c}_k)$ to an allocation $n(\hat{c}) \in \mathbb{R}_{\geq 0}^k$ and payments $p(\hat{c}) \in \mathbb{R}_{\geq 0}^k$ (where $p_i$ is paid *to* seller $i$).

**Utilities and feasibility.** Seller $i$ is quasi-linear with utility

$$u_i(\hat{c}; c_i) = p_i(\hat{c}) - c_i \, n_i(\hat{c}).$$

The buyer's (computational) acquisition problem induces an allocation objective $U(n)$.

*Induced-cost feasibility* requires that for every report profile $\hat{c}$,

$$\sum_{i=1}^{k} \hat{c}_i \, n_i(\hat{c}) \leq B \quad \text{and} \quad n_i(\hat{c}) \geq 0 \; \forall i.$$

Under truthful reports (which we guarantee below), this coincides with the true induced-cost constraint $\sum_i c_i n_i(c) \leq B$. We also impose *no positive transfers to the buyer*: $p_i(\hat{c}) \geq 0$ for all $i, \hat{c}$.

**Truthfulness and individual rationality.** A mechanism is *dominant-strategy incentive compatible* (DSIC) if for every $i$ and all $c_i, \hat{c}_i, \hat{c}_{-i}$,

$$p_i(\hat{c}_i, \hat{c}_{-i}) - c_i \, n_i(\hat{c}_i, \hat{c}_{-i}) \geq p_i(c_i, \hat{c}_{-i}) - c_i \, n_i(c_i, \hat{c}_{-i}).$$

It is *ex-post individually rational* (IR) if $p_i(c) - c_i n_i(c) \geq 0$ for all $c$.

## A.1. Monotonicity and Payment Characterizations

Each seller is characterized by a single-parameter (per-unit cost). The buyer purchases a divisible quantity $n_i(\cdot) \in \mathbb{R}_{\geq 0}$. The canonical Myerson characterization specializes as follows.

**Theorem 2** (DSIC $\Leftrightarrow$ monotonicity + area-under-the-curve payments). *Fix any seller $i$. A direct mechanism $(n, p)$ is DSIC and ex-post IR for seller $i$ if and only if:*

1. **Monotonicity in own cost:** *for every $\hat{c}_{-i}$, the allocation $\hat{c}_i \mapsto n_i(\hat{c}_i, \hat{c}_{-i})$ is weakly non-increasing.*

2. **Payment identity (integral form):** *for every $c_i$ and $\hat{c}_{-i}$,*

$$p_i(c_i, \hat{c}_{-i}) = c_i \, n_i(c_i, \hat{c}_{-i}) + \int_{c_i}^{\overline{c}_i} n_i(t, \hat{c}_{-i}) \, dt.$$

*Under* randomized *allocations (e.g., from stochastic oracles), DSIC and IR hold* in expectation *when conditions (1)–(2) hold for the interim allocation $\mathbb{E}[n_i(\cdot)]$ and payments $\mathbb{E}[p_i(\cdot)]$.*

## A.2. Implementing Acquisition Algorithms as Mechanisms

Let $\mathcal{A}$ be a (possibly randomized) acquisition algorithm that, given reported costs $\hat{c}$, a compute budget $C$, and access to attribution oracles, returns a quantity vector $\tilde{n}(\hat{c})$ that (approximately) maximizes the buyer's utility $U(n)$ subject to the induced-cost constraint $\sum_i \hat{c}_i n_i \leq B$.

To convert $\mathcal{A}$ into a truthful mechanism:

1. **Commitment.** The buyer publicly commits to the algorithmic map $\hat{c} \mapsto \tilde{n}(\hat{c})$ (including oracle schedules, step sizes, stopping rules) and to truthful payments defined above.

2. **Monotonicity check.** If $\tilde{n}$ is *monotone in each coordinate* (weakly non-increasing in $\hat{c}_i$ for fixed $\hat{c}_{-i}$), then by Theorem 2 the payment identity makes the mechanism DSIC and ex-post IR (truthful reports $c = \hat{c}$ restore true induced-cost feasibility).

### A.3. Approximate Solutions and $\varepsilon$-Truthfulness

**Definition 3** ($\eta$-monotonicity). *An allocation $\tilde{n}$ is $\eta$-monotone if for all $i$ and all $\hat{c}_1 < \hat{c}_2$,*

$$\tilde{n}_i(\hat{c}_2, \hat{c}_{-i}) \ \leq \ \tilde{n}_i(\hat{c}_1, \hat{c}_{-i}) \ + \ \eta.$$

**Lemma 1** (Approximate DSIC). *Suppose $\tilde{n}$ is $\eta$-monotone and payments use Myerson's integral identity computed on $\tilde{n}$. Then truthful reporting is a $(\eta \cdot (\bar{c}_i - \underline{c}_i))$-approximate dominant strategy for seller $i$. In particular, no seller can gain more than $O(\eta \Delta c)$ by misreporting. If $\eta \to 0$ as the compute budget $C$ grows, the mechanism converges to DSIC.*

Interestingly, there are ways to restore monotonicity and obtain *exact* truthful mechanisms by using *ironing*. This is outside the scope of our work; for details we refer the reader to Toikka (2011); Roughgarden & Schrijvers (2016).

### A.4. Randomization and Truthfulness in Expectation

If $\mathcal{A}$ uses randomized oracles (e.g., stochastic trace estimators) then $n(\hat{c})$ is random. Define the interim allocation $\bar{n}(\hat{c}) = \mathbb{E}[n(\hat{c})]$ and interim payments $\bar{p}(\hat{c}) = \mathbb{E}[p(\hat{c})]$. If $\bar{n}_i(\cdot)$ is non-increasing in $\hat{c}_i$, then the mechanism is truthful *in expectation* (DSIC-in-expectation) and ex-post IR in expectation, using the payment formula we described above.

### A.5. Induced-Cost vs. Budget-Feasible Procurement

Classical budget-feasible mechanisms constrain the buyer's *total payment*. Our setting constrains the *true induced cost* $\sum_i c_i n_i$ (annotation hours, API caps, compliance overhead). Under DSIC, reports are truthful and the reported-cost feasibility coincides with true feasibility. Payments remain the instrument achieving participation and truthfulness; they do not determine feasibility. This separation lets the buyer optimize compute scheduling (oracle fidelity across iterations) independently of incentive constraints, as long as the resulting allocation rule is monotone in each $c_i$.

**Takeaway.** To turn any oracle-driven acquisition pipeline into a truthful mechanism under an induced-cost cap: (i) ensure (or enforce by "ironing") monotonicity of $n_i(\cdot)$ in $c_i$; (ii) compute the payments via Myerson's formula; (iii) compute feasibility using reported costs—truthfulness.

## B. Omitted Details from Section 3

### B.1. Preliminaries

In this section, we provide a few useful lemmata that will be used in the proof of the main results.

**Lemma 2** (Precision matrix concentration, (Kereta & Klock, 2021)). *Let $X$ be a subgaussian random variable with mean zero and subgaussian parameter $\sigma$, let $X_1, \ldots, X_n$ be i.i.d. copies of $X$, let $\Sigma = \mathbb{E}[XX^\top]$, and let $\hat{\Sigma}_n = \frac{1}{n} \sum_{i=1}^n X_i X_i^\top$ be the sample covariance matrix. Then, for any $\delta > 0$,*

$$\mathbb{P}\left[ \left\| \hat{\Sigma}_n^{-1} - \Sigma^{-1} \right\|_F^2 \geq \frac{\sigma^2}{\lambda_{\min}(\Sigma)^2} \sqrt{\frac{d + \log(1/\delta)}{n}} \right] \leq \delta.$$

**Lemma 3** (Convergence of approximate gradient descent (Schmidt et al., 2011)). *Let $f$ be a convex function with an $L$-Lipschitz gradient, and let $S$ be a convex set. Consider the projected gradient descent update rule*

$$x_{t+1} = \Pi_S \left( x_t - \frac{1}{L} \left[ \nabla f(x_t) + e_t \right] \right).$$

*The average iterate $\bar{x}_t = \sum_{i=1}^t x_t$ satisfies*

$$f(\bar{x}_t) - f(x^*) \leq \frac{L}{2t} \left( \operatorname{diam}(S) + \frac{2}{L} \sum_{i=1}^t \|e_i\|_2 \right),$$

*where $x^* = \arg\min_{x \in S} f(x)$.*

The next is a standard result in probability theory.

**Lemma 4** (Conditional Distribution of Independent Poisson Variables (Casella & Berger, 2024)). *Let $X_1, \ldots, X_k$ be independent Poisson random variables with parameters $\lambda_1, \ldots, \lambda_k$. Then, the conditional distribution of $(X_1, \ldots, X_k)$ given $\sum_{i=1}^{k} X_i = n$ is multinomial with parameters $n$ and $\left( \frac{\lambda_1}{\sum_{j=1}^{k} \lambda_j}, \ldots, \frac{\lambda_k}{\sum_{j=1}^{k} \lambda_j} \right)$. Furthermore, the conditional distribution of $\sum_{i=1}^{k} X_i$ is a Poisson random variable with parameter $\sum_{i=1}^{k} \lambda_i$.*

### B.2. Gradient Derivation

$$U(\pi) = \sum_{\vec{n} \in \mathbb{N}^k} p_\pi(\vec{n}) \, g(\vec{n}), \qquad \text{where} \qquad g(\vec{n}) := \mathbb{E}\big[\mathrm{tr}\big((X^\top X)^{-1}\big) \,\big|\, \vec{n}\big]$$

$$
\begin{aligned}
\frac{\partial}{\partial \pi_i} U(\pi) &= \sum_{\vec{n}} g(\vec{n}) \, \frac{\partial}{\partial \pi_i} p_\pi(\vec{n}) \\
&= \sum_{\vec{n}} g(\vec{n}) \left( p_\pi(\vec{n} - e_i) - p_\pi(\vec{n}) \right) \\
&= \sum_{\vec{n}} \left( g(\vec{n} + e_i) - g(\vec{n}) \right) p_\pi(\vec{n}) \\
&= \mathbb{E}_{\vec{n} \sim \mathrm{Poisson}(\pi)} [g(\vec{n} + e_i) - g(\vec{n})] \\
&= \mathbb{E}_{\substack{\vec{n} \sim \mathrm{Poisson}(\pi) \\ X \mid \vec{n}, \, x \sim \mathcal{D}_i}} \left[ \mathrm{tr}\big((X^\top X + xx^\top)^{-1}\big) - \mathrm{tr}\big((X^\top X)^{-1}\big) \right] \\
&= - \mathbb{E}_{\substack{\vec{n} \sim \mathrm{Poisson}(\pi) \\ X \mid \vec{n}, \, x \sim \mathcal{D}_i}} \left[ \frac{\|(X^\top X)^{-1} x\|_2^2}{1 + x^\top (X^\top X)^{-1} x} \right]
\end{aligned}
$$

where $p_\pi(\vec{n}) = \prod_{\ell=1}^{k} e^{-\pi_\ell} \pi_\ell^{n_\ell} / n_\ell!$, $e_i$ is the $i$-th standard basis vector (with the convention $p_\pi(\vec{n} - e_i) = 0$ when $n_i = 0$), and $X$ is the design matrix formed by stacking the sampled covariates. The third line uses the identity $\frac{\partial}{\partial \lambda}\big(e^{-\lambda} \lambda^n / n!\big) = e^{-\lambda} \lambda^{n-1} / (n-1)! - e^{-\lambda} \lambda^n / n!$, and the last line follows from the Sherman–Morrison formula applied to $X^\top X + xx^\top$.

### B.3. Proof of Proposition 1

**Proposition 6** (Large-budget correctness of Algorithm 1). *Fix $\pi \in \mathbb{R}_{\geq 0}^k$ and Let $\widehat{g}(\pi, \varepsilon) \in \mathbb{R}^k$ be the output of Algorithm 1, so that $\big|\widehat{g}_i(\pi, \varepsilon) + \mathrm{tr}(\Sigma_i S^{-2})\big| \leq \varepsilon$ for each $i \in [k]$. Assume the subgaussian and spectral conditions in Section 3.1 and fix $\delta \in (0, 1/2)$. If*

$$B \geq c_{max} \cdot \max \left\{ 12 \log \frac{4}{\delta}, \ 2\sigma_x^4 \Big(d + \log \frac{4}{\delta}\Big) \right\},$$

*then for every $i \in [k]$ the oracle output is close to the true Poissonized gradient* (1):

$$\big|\widehat{g}_i(\pi, \varepsilon) - \nabla_i f(\pi)\big| \leq \varepsilon + \frac{1024 \, \sigma_x^4 d^2}{m^3 B^3} + \frac{32 M d}{m^2 B^2}(1 + \delta).$$

*Proof.* We will show that

$$\left| \frac{1}{B} \widehat{g}_i(\pi, \varepsilon) - \frac{1}{B} \nabla_i f(\pi) \right| \leq \frac{1}{B}\varepsilon + \frac{25 M d}{m^2 B^2} + \frac{1024 \, \sigma_x^4 d^2}{m^3 B^3} + \frac{32 M d}{m^2 B^2} \delta.$$

Fix some $i \in [k]$. Let $\vec{n} \sim \mathrm{Poisson}(\pi)$, and let $N := \|\vec{n}\|_1$ and $\widetilde{N} := \sum_{i=1}^{k} \pi_i$, so that $N \sim \mathrm{Poisson}(\widetilde{N})$. Observe that $\widetilde{N} \geq \frac{B}{c_{max}}$, and so our assumption on $B$ implies a lower bound on $\widetilde{N}$. Let $X$ be the pooled design matrix sampled according to $\vec{n}$, and let $A := X^\top X$.

For $x \sim \mathcal{D}_i$ independent of $(\vec{n}, X)$, define $\phi_i := \frac{\|A^{-1} x\|_2^2}{1 + x^\top A^{-1} x}$, so $\nabla_i f(\pi) = -\mathbb{E}[\phi_i]$ by (1). Define

$$\Sigma_\pi := \frac{1}{\widetilde{N}} \sum_{j=1}^{k} \pi_j \Sigma_j \qquad \text{and} \qquad \hat{\Sigma} := \frac{1}{N} X^\top X \cdot \mathbf{1}\{N \geq 1\}.$$

Note that $mI \preceq \Sigma_\pi \preceq MI$ and, conditional on $N$, the pooled covariates are i.i.d. $\sigma_x$-subgaussian with covariance $\Sigma_\pi$ (by Lemma 4). Let $Z := \|\hat{\Sigma}^{-1} - \Sigma_\pi^{-1}\|_F^2$ and define the event

$$E := \left\{ N \geq \frac{\widetilde{N}}{2} \right\} \cap \left\{ Z \leq \frac{\sigma_x^2}{m^2} \sqrt{\frac{2(d + \log(4/\delta))}{\widetilde{N}}} \right\}.$$

A Poisson tail bound gives $\mathbb{P}[N < \widetilde{N}/2] \leq \exp(-\widetilde{N}/12) \leq \delta/4$ as long as $\widetilde{N} \geq 12 \log(4/\delta)$.

Next using Lemma 4, we can apply Lemma 2 conditional on $N$, to the mixture distribution with covariance $\Sigma_\pi$. On the event $\{N \geq \widetilde{N}/2\}$ we have

$$\mathbb{P}\left( Z > \frac{\sigma_x^2}{m^2} \sqrt{\frac{2(d + \log(4/\delta))}{\widetilde{N}}} \mid N \right) \leq \delta,$$

hence $\mathbb{P}(E^c) \leq \mathbb{P}(N < \widetilde{N}/2) + \delta \leq 5\delta/4$. On $E$, we have $\|\hat{\Sigma}^{-1}\|_2 \leq \|\Sigma_\pi^{-1}\|_2 + \sqrt{Z} \leq 1/m + 1/m = 2/m$, so $\frac{1}{N}\|\hat{\Sigma}^{-1}\|_2 \leq 4/(m\widetilde{N})$.

**Step 1: The denominator is negligible.** Since $t/(1+t) \leq t$ for all $t \geq 0$,

$$0 \leq \mathbb{E}_x\big[x^\top A^{-2} x\big] - \mathbb{E}_x[\phi_i] = \mathbb{E}_x\left[ (x^\top A^{-2} x) \frac{x^\top A^{-1} x}{1 + x^\top A^{-1} x} \right] \leq \mathbb{E}_x\big[(x^\top A^{-2} x)(x^\top A^{-1} x)\big] \leq \|A^{-1}\|_2^3 \, \mathbb{E}\|x\|_2^4.$$

Because $x$ is $\sigma_x$-subgaussian, each coordinate has fourth moment at most $16\sigma_x^4$, so $\mathbb{E}\|x\|_2^4 \leq 16\sigma_x^4 d^2$; on $E$ the displayed quantity is therefore at most $1024\,\sigma_x^4 d^2/(m^3 B^3)$.

**Step 2: The numerator concentrates.** We have $\mathbb{E}_x[x^\top A^{-2} x] = \text{tr}(\Sigma_i A^{-2})$ and

$$\big|\text{tr}(\Sigma_i A^{-2}) - \text{tr}(\Sigma_i S^{-2})\big| \leq \|\Sigma_i\|_F \|A^{-2} - S^{-2}\|_F \leq \sqrt{d}\, M \left(\|A^{-1}\|_2 + \|S^{-1}\|_2\right) \|A^{-1} - S^{-1}\|_F,$$

using $A^{-2} - S^{-2} = A^{-1}(A^{-1} - S^{-1}) + (A^{-1} - S^{-1})S^{-1}$. On $E$, $\|S^{-1}\|_2 \leq 1/(mB)$ and $\|A^{-1}\|_2 \leq 4/(mB)$. Also

$$A^{-1} - S^{-1} = \frac{1}{N}(\hat{\Sigma}^{-1} - \Sigma_\pi^{-1}) + \left(\frac{1}{N} - \frac{1}{B}\right)\Sigma_\pi^{-1},$$

so on $E$ we have $\|A^{-1} - S^{-1}\|_F \leq \frac{2}{B}\sqrt{Z} + \frac{3}{B}\|\Sigma_\pi^{-1}\|_F \leq \frac{5\sqrt{d}}{mB}$, giving $\big|\text{tr}(\Sigma_i A^{-2}) - \text{tr}(\Sigma_i S^{-2})\big| \leq 25Md/(m^2 B^2)$.

Combining Step 1 and Step 2 and conditioning on $E$, we get the bound

$$\big|\mathbb{E}_x[\phi_i] - \text{tr}(\Sigma_i S^{-2})\big| \leq 25Md/(m^2 B^2) + 1024\,\sigma_x^4 d^2/(m^3 B^3).$$

Finally, since $\mathbb{P}(E^c) \leq 5\delta/4$ and $\text{tr}(\Sigma_i S^{-2}) \leq Md/(m^2 B^2)$, taking expectations and using $\big|\hat{g}_i(\pi, \varepsilon) - \text{tr}(\Sigma_i S^{-2})\big| \leq \varepsilon$ gives the claimed inequality. $\quad\square$

### B.4. Proof of Proposition 2

*Proof.* Our proof proceeds in three steps. We first show that the (deterministic) $A$-optimal design surrogate is convex; we then bound the bias of the oracle-induced gradient, allowing us to invoke Lemma 3; finally, we relate the optimum of the $A$-optimal design surrogate back to the Poissonized objective via concentration bounds.

**Step 1: $A$-optimal design is convex and $L$-smooth.** Recall our notation from Section 3.1, the $A$-optimal design objective is given by

$$\hat{\ell}(\pi) := B \cdot \text{tr}\big(S(\pi)^{-1}\big).$$

We consider optimizing this objective over the convex feasible set $\{\pi \in \mathbb{R}_{\geq 0}^k : \sum_i c_i \pi_i \leq B\}$. Since $\pi \mapsto S(\pi)$ is affine and $S \mapsto \text{tr}(S^{-1})$ is convex on the positive-definite cone (Pukelsheim, 2006), it follows that $\hat{\ell}$ is convex in $\pi$.

To

**Step 2: Apply our approximate-gradient lemma.** From Proposition 1, we have that the output of Algorithm 1 is a $\varepsilon$-approximate oracle for $\widehat{\ell}(\pi)$. By applying Lemma 3, we have that, after $t$ gradient updates, the average iterate $\overline{\pi_t}$ satisfies

$$\widehat{\ell}(\overline{\pi_t}) - \widehat{\ell}(\pi^*) \le \frac{L}{2t}\left(\operatorname{diam}(\Delta_k) + \frac{2}{L}\sum_{s=1}^{t}\|\varepsilon_s\|_2\right), \tag{2}$$

where $\pi^*$ is the minimizer of $\widehat{\ell}(\pi)$ over the budget-feasible set.

**Step 3: Relate $A$-optimal design to Poissonized objective.** Recall that our sampling process is given by $n_i \sim \operatorname{Poisson}(\frac{B\pi_i}{c_i})$. Applying Lemma 4, we can rewrite our original objective as

$$\ell(\pi) = B \cdot \mathbb{E}_{N\sim\operatorname{Poisson}(\sum_{i=1}^{k}\pi_i)}\left[\mathbb{E}_{\vec{n}\sim\operatorname{Multinomial}(N,\frac{\pi_1}{\sum_{i=1}^{k}\pi_i},\ldots,\frac{\pi_k}{\sum_{i=1}^{k}\pi_i})}\left[\operatorname{tr}(\widehat{S}(\vec{n})^{-1})\right]\right]$$

$$= B \cdot \mathbb{E}_{N\sim\operatorname{Poisson}(\sum_{i=1}^{k}\pi_i)}\left[\operatorname{tr}\left(\mathbb{E}_{\vec{n}\sim\operatorname{Multinomial}(N,\frac{\pi_1}{\sum_{i=1}^{k}\pi_i},\ldots,\frac{\pi_k}{\sum_{i=1}^{k}\pi_i})}\left[\widehat{S}(\vec{n})^{-1}\right]\right)\right]$$

$$= \mathbb{E}_{N\sim\operatorname{Poisson}(\sum_{i=1}^{k}\pi_i)}\left[\frac{B}{N}\cdot\operatorname{tr}\left(\mathbb{E}_{\vec{n}\sim\operatorname{Multinomial}(N,\frac{\pi_1}{\sum_{i=1}^{k}\pi_i},\ldots,\frac{\pi_k}{\sum_{i=1}^{k}\pi_i})}\left[\left(\frac{1}{N}\widehat{S}(\vec{n})\right)^{-1}\right]\right)\right].$$

By Lemma 2, we have that, with probability at least $1-\delta$,

$$\left\|\mathbb{E}_{\vec{n}\sim\operatorname{Multinomial}(N,\frac{\pi_1}{\sum_{i=1}^{k}\pi_i},\ldots,\frac{\pi_k}{\sum_{i=1}^{k}\pi_i})}\left[\left(\frac{1}{N}\widehat{S}(\vec{n})\right)^{-1}\right] - \left(\sum_{i=1}^{k}\frac{\pi_i}{\sum_{j=1}^{k}\pi_j}\Sigma_i\right)^{-1}\right\|_F^2 \le \frac{\sigma^2}{m^2}\sqrt{\frac{d+\log(1/\delta)}{N}}.$$

Since $N \in \Theta(B)$, we have that, with probability at least $1-\delta$ this error converges to zero as $B \to \infty$, and thus $\ell(\pi_T)$ converges to $\widehat{\ell}(\pi_T)$ (see the proof of Proposition 1 in Appendix B.3 for the details). Finally, we can use (2) to relate $\ell(\overline{\pi_T})$ to $\ell(\pi^*)$, concluding the proof. $\square$

### B.5. Proof of Proposition 3

*Proof.* For convenience, let $S_0 = S(\pi_0)$, and define $s_i := \operatorname{tr}(S_0^{-1}\Sigma_i S_0^{-1})$ so that $\nabla\widehat{f}(\pi_0) = -(s_1,\ldots,s_k)$. Moreover,

$$\bar{s} = \frac{1}{k}\sum_{i=1}^{k}\operatorname{tr}(S_0^{-1}\Sigma_i S_0^{-1}) = \operatorname{tr}\left(S_0^{-1}\left(\frac{1}{k}\sum_{i=1}^{k}\Sigma_i\right)S_0^{-1}\right) = \operatorname{tr}(S_0^{-1}S_0 S_0^{-1}) = \operatorname{tr}(S_0^{-1}).$$

Since feasible directions satisfy $\mathbf{1}^\top\pi = 1$, the affine-feasible (sum-zero) component of the descent direction $-\nabla U_A(\pi_0) = s$ is its centered version $v = s - \bar{s}\mathbf{1}$, which satisfies $\mathbf{1}^\top v = 0$. The Euclidean projection of $\pi_0 + \eta s$ onto the affine hyperplane $\{\pi : \mathbf{1}^\top\pi = 1\}$ is therefore $\pi_0 + \eta v$; under the interior-step assumption (no coordinate clipping), the simplex projection agrees, so the first projected-gradient iterate is $\pi_1 = \pi_0 + \eta v$.

Along the line segment $\pi(t) = \pi_0 + t(\pi_1 - \pi_0) = \pi_0 + t\eta v$ (which stays feasible by convexity), the information matrix evolves as

$$S(t) := S(\pi(t)) = S_0 + t\eta H, \qquad H := \sum_{i=1}^{k}v_i\Sigma_i,$$

and we define $\phi(t) := \operatorname{tr}(S(t)^{-1})$, so that $U_A(\pi_0) = \phi(0)$ and $U_A(\pi_1) = \phi(1)$. Differentiating gives

$$\phi'(t) = -\eta\operatorname{tr}\left(S(t)^{-1}HS(t)^{-1}\right),$$

hence, at $t = 0$,

$$\phi'(0) = -\eta\operatorname{tr}\left(S_0^{-1}HS_0^{-1}\right) = -\eta\sum_{i=1}^{k}v_i\operatorname{tr}\left(S_0^{-1}\Sigma_i S_0^{-1}\right)$$

$$= -\eta\sum_{i=1}^{k}v_i s_i = -\eta\sum_{i=1}^{k}(s_i-\bar{s})s_i = -\eta\sum_{i=1}^{k}(s_i-\bar{s})^2 = -\eta\|v\|_2^2.$$

A second differentiation yields

$$\phi''(t) = 2\eta^2 \operatorname{tr}\big(S(t)^{-1} H S(t)^{-1} H S(t)^{-1}\big) = 2\eta^2 \left\|S(t)^{-1} H S(t)^{-1/2}\right\|_F^2.$$

Because $\pi(t)$ is feasible and $mI \preceq \Sigma_i \preceq MI$, we have $mI \preceq S(t) \preceq MI$ for all $t \in [0, 1]$, hence $\|S(t)^{-1}\|_2 \leq 1/m$, $\|S(t)^{-1/2}\|_2 \leq 1/\sqrt{m}$ and also $\sigma_{\min}(S(t)^{-1}) \geq 1/M$, $\sigma_{\min}(S(t)^{-1/2}) \geq 1/\sqrt{M}$. Using $\|BXC\|_F \leq \|B\|_2 \|C\|_2 \|X\|_F$ and $\|BXC\|_F \geq \sigma_{\min}(B)\sigma_{\min}(C)\|X\|_F$, we obtain

$$\frac{1}{M^{3/2}} \|H\|_F \leq \left\|S(t)^{-1} H S(t)^{-1/2}\right\|_F \leq \frac{1}{m^{3/2}} \|H\|_F,$$

and therefore for all $t \in [0, 1]$,

$$\frac{2\eta^2}{M^3} \|H\|_F^2 \leq \phi''(t) \leq \frac{2\eta^2}{m^3} \|H\|_F^2.$$

Taylor's theorem with integral remainder gives

$$\phi(1) = \phi(0) + \phi'(0) + \int_0^1 (1 - t)\phi''(t)\, dt,$$

so rearranging and using $\int_0^1 (1 - t)\, dt = \frac{1}{2}$ yields

$$U_A(\pi_0) - U_A(\pi_1) = \phi(0) - \phi(1) = -\phi'(0) - \int_0^1 (1 - t)\phi''(t)\, dt \leq \eta\|v\|_2^2 - \frac{\eta^2}{M^3} \|H\|_F^2.$$

If $\lambda \geq \eta\|v\|_2^2 - \frac{\eta^2}{M^3} \|H\|_F^2$, then $U_A(\pi_0) - U_A(\pi_1) \leq \lambda$, equivalently $U_A(\pi_1) + \lambda \geq U_A(\pi_0)$. Thus paying for this gradient query cannot improve the buyer's objective, so the compute-optimal policy is to stop at the uniform allocation $\pi_0$. $\qquad \square$

### B.6. Proof of Proposition 4

*Proof.* Write $g_t := \nabla \widehat{f}(\pi_t)$ and recall $\pi_{t+1} = \pi_t - \eta g_t$. Define $\Delta_t := \widehat{f}(\pi_t) - \widehat{f}(\pi_{t+1})$, so $J_{t+1} - J_t = \lambda - \Delta_t$.

*Step 1: $\Delta_t$ is nonincreasing.* We use the standard cocoercivity inequality (Baillon–Haddad): for convex $L$-smooth $\widehat{f}$,

$$\langle \nabla \widehat{f}(x) - \nabla \widehat{f}(y),\, x - y \rangle \geq \frac{1}{L} \left\|\nabla \widehat{f}(x) - \nabla \widehat{f}(y)\right\|_2^2 \qquad \forall x, y. \tag{3}$$

Convexity also gives the supporting-hyperplane inequality $\widehat{f}(u) \geq \widehat{f}(v) + \langle \nabla \widehat{f}(v), u - v \rangle$. Applying it with $(u, v) = (\pi_{t+1}, \pi_t)$ yields

$$\widehat{f}(\pi_{t+1}) \geq \widehat{f}(\pi_t) + \langle g_t, \pi_{t+1} - \pi_t \rangle = \widehat{f}(\pi_t) - \eta\|g_t\|_2^2,$$

hence $\Delta_t \leq \eta\|g_t\|_2^2$. Applying it with $(u, v) = (\pi_t, \pi_{t+1})$ yields

$$\widehat{f}(\pi_t) \geq \widehat{f}(\pi_{t+1}) + \langle g_{t+1}, \pi_t - \pi_{t+1} \rangle = \widehat{f}(\pi_{t+1}) + \eta\langle g_{t+1}, g_t \rangle,$$

hence $\Delta_t \geq \eta\langle g_{t+1}, g_t \rangle$.

Next apply (3) with $(x, y) = (\pi_t, \pi_{t+1})$. Since $\pi_t - \pi_{t+1} = \eta g_t$, we obtain

$$\eta\langle g_t - g_{t+1}, g_t \rangle \geq \frac{1}{L} \|g_t - g_{t+1}\|_2^2.$$

Dividing by $\eta > 0$ and using $\eta \leq 1/L$ gives

$$\langle g_t - g_{t+1}, g_t \rangle \geq \frac{1}{L\eta} \|g_t - g_{t+1}\|_2^2 \geq \|g_t - g_{t+1}\|_2^2.$$

Expanding both sides implies $\langle g_{t+1}, g_t \rangle \geq \|g_{t+1}\|_2^2$. Therefore,

$$\Delta_t \geq \eta\langle g_{t+1}, g_t \rangle \geq \eta\|g_{t+1}\|_2^2 \qquad \text{and} \qquad \Delta_{t+1} \leq \eta\|g_{t+1}\|_2^2,$$

so $\Delta_{t+1} \leq \Delta_t$ as claimed.

*Step 2: one-step-overhead optimal stopping.* Observe that

$$J_{t+1} - J_t \; = \; \big(\widehat{f}(\pi_{t+1}) - \widehat{f}(\pi_t)\big) + \lambda \; = \; \lambda - \Delta_t.$$

Since $(\Delta_t)$ is nonincreasing, the increments $(J_{t+1} - J_t)$ are nondecreasing in $t$. Hence $J_t$ decreases as long as $\Delta_t > \lambda$ and (once $\Delta_t \leq \lambda$) can never decrease again. Let $\hat{t}$ be the first index with $J_{\hat{t}+1} \geq J_{\hat{t}}$. Then $J_t$ is nonincreasing on $\{0, 1, \ldots, \hat{t}\}$ and nondecreasing on $\{\hat{t}, \hat{t}+1, \ldots\}$, so $J_{\hat{t}} = \min_{t \geq 0} J_t$. In particular, $\hat{t}$ is the earliest minimizer, hence $\hat{t} = T^\star$ by definition. Finally, the rule can only detect $J_{\hat{t}+1} \geq J_{\hat{t}}$ after forming $\pi_{\hat{t}+1}$, so it uses $\hat{t} + 1 = T^\star + 1$ gradient evaluations. $\qquad\square$

# C. Results on Compute Schedule

*Proof of Proposition 5.* Let $e_t := \tau^{(t)} - \nabla U_A(\pi_t)$, so that $\|e_t\|_2 \leq \epsilon_t$. Since $\pi^\star$ minimizes $U_A$ over the closed convex set $\Delta_k$, it satisfies the fixed-point condition

$$\pi^\star = \Pi_{\Delta_k}\big(\pi^\star - \eta \nabla U_A(\pi^\star)\big).$$

By nonexpansiveness of Euclidean projection,

$$\|\pi_{t+1} - \pi^\star\|_2 = \left\| \Pi_{\Delta_k}\big(\pi_t - \eta(\nabla U_A(\pi_t) + e_t)\big) - \Pi_{\Delta_k}\big(\pi^\star - \eta \nabla U_A(\pi^\star)\big) \right\|_2$$
$$\leq \|\pi_t - \pi^\star - \eta(\nabla U_A(\pi_t) - \nabla U_A(\pi^\star))\|_2 \; + \; \eta \|e_t\|_2.$$

Because $U_A$ is $C^2$ and $\mu$-strongly convex / $L$-smooth on $\Delta_k$, we have $\mu I \preceq \nabla^2 U_A(\xi) \preceq LI$ for all $\xi \in \Delta_k$. By the mean value theorem,

$$\nabla U_A(\pi_t) - \nabla U_A(\pi^\star) = J_t(\pi_t - \pi^\star) \quad \text{for some symmetric } J_t \text{ with } \mu I \preceq J_t \preceq LI.$$

Therefore,

$$\|\pi_t - \pi^\star - \eta(\nabla U_A(\pi_t) - \nabla U_A(\pi^\star))\|_2 = \|(I - \eta J_t)(\pi_t - \pi^\star)\|_2 \leq \|I - \eta J_t\|_2 \, \|\pi_t - \pi^\star\|_2.$$

Since $\eta \leq 1/L$ and the eigenvalues of $J_t$ are in $[\mu, L]$, the eigenvalues of $I - \eta J_t$ lie in $[0, 1 - \mu\eta]$, hence $\|I - \eta J_t\|_2 \leq 1 - \mu\eta$. Let $\rho := 1 - \mu\eta \in [0, 1)$. Using $\|e_t\|_2 \leq \epsilon_t$ gives the recursion

$$\|\pi_{t+1} - \pi^\star\|_2 \; \leq \; \rho \|\pi_t - \pi^\star\|_2 + \eta \epsilon_t.$$

Unrolling yields the standard last-iterate bound

$$\|\pi_T - \pi^\star\|_2 \; \leq \; \rho^T \|\pi_0 - \pi^\star\|_2 \; + \; \sum_{t=0}^{T-1} w_t \epsilon_t, \qquad w_t := \eta \rho^{T-1-t}.$$

Since $0 \leq \rho < 1$, we have $0 < w_0 < w_1 < \cdots < w_{T-1}$ (strictly when $\rho \in (0, 1)$).

Thus minimizing $\mathcal{J}$ over schedules is equivalent to minimizing

$$\widetilde{\mathcal{J}}(\epsilon_0, \ldots, \epsilon_{T-1}) \; := \; \sum_{t=0}^{T-1} w_t \epsilon_t \; + \; \lambda \sum_{t=0}^{T-1} \gamma(\epsilon_t),$$

since the term $\rho^T \|\pi_0 - \pi^\star\|_2$ is constant w.r.t. the schedule.

Let $\epsilon^\star = (\epsilon_0^\star, \ldots, \epsilon_{T-1}^\star)$ be an optimizer of $\widetilde{\mathcal{J}}$. Suppose for contradiction that there exist indices $s < t$ such that $\epsilon_s^\star < \epsilon_t^\star$. Define $\widetilde{\epsilon}$ by swapping the values at coordinates $s$ and $t$ and leaving all other coordinates unchanged. Then the compute term is unchanged because it is a sum:

$$\sum_{u=0}^{T-1} \gamma(\widetilde{\epsilon}_u) = \sum_{u=0}^{T-1} \gamma(\epsilon_u^\star).$$

The weighted error term changes by

$$\left(w_s\widetilde{\epsilon}_s + w_t\widetilde{\epsilon}_t\right) - \left(w_s\epsilon_s^\star + w_t\epsilon_t^\star\right) = \left(w_s\epsilon_t^\star + w_t\epsilon_s^\star\right) - \left(w_s\epsilon_s^\star + w_t\epsilon_t^\star\right)$$
$$= (w_t - w_s)(\epsilon_s^\star - \epsilon_t^\star) < 0,$$

since $w_t > w_s$ and $\epsilon_s^\star - \epsilon_t^\star < 0$. Hence $\widetilde{\mathcal{J}}(\widetilde{\epsilon}) < \widetilde{\mathcal{J}}(\epsilon^\star)$, contradicting optimality. Therefore no such inversion exists and $\epsilon_0^\star \geq \epsilon_1^\star \geq \cdots \geq \epsilon_{T-1}^\star$.

Finally, if $\gamma$ is nonincreasing, then $\epsilon_s^\star \geq \epsilon_t^\star$ implies $\gamma(\epsilon_s^\star) \leq \gamma(\epsilon_t^\star)$, so the compute costs are nondecreasing in time (back-loaded). $\square$

**Corollary 1** (Closed-form schedule for power-law oracle costs). *Assume the oracle cost takes the form $\gamma(\epsilon) = a\,\epsilon^{-p}$ for some $a > 0$ and $p > 0$. In the compute-aware objective*

$$\sum_{t=0}^{T-1} w_t\epsilon_t + \lambda \sum_{t=0}^{T-1} \gamma(\epsilon_t), \qquad w_t = \eta\rho^{T-1-t},$$

*an optimizer is given by the per-iteration minimizers*

$$\epsilon_t^\star = \left(\frac{\lambda a p}{w_t}\right)^{\frac{1}{p+1}} = \left(\frac{\lambda a p}{\eta}\right)^{\frac{1}{p+1}} \rho^{-\frac{T-1-t}{p+1}}.$$

*In particular, the schedule is geometric:*

$$\epsilon_{t+1}^\star = \rho^{\frac{1}{p+1}} \epsilon_t^\star, \qquad so \qquad \epsilon_0^\star \geq \epsilon_1^\star \geq \cdots \geq \epsilon_{T-1}^\star.$$

*Moreover, the per-iteration compute spends are also geometric and back-loaded: $\gamma(\epsilon_{t+1}^\star) = \rho^{-\frac{p}{p+1}}\gamma(\epsilon_t^\star)$.*

*Proof.* Because the objective is a sum of independent terms $w_t\epsilon_t + \lambda a\epsilon_t^{-p}$, each $\epsilon_t^\star$ is obtained by minimizing this one-dimensional function over $\epsilon_t > 0$. Differentiating and setting to zero gives $w_t - \lambda a p\,\epsilon_t^{-(p+1)} = 0$, hence the stated formula. The geometric ratio follows from $w_{t+1} = w_t/\rho$. $\square$

# D. Omitted Results from Section 3.4

## D.1. Monotonicity of Optimal Solution

*Proof of Theorem 1.* Write $\Delta := c_j^{(2)} - c_j^{(1)} \geq 0$. It is not hard to see that both optima spend the full budget:

$$\langle c^{(1)}, w^{(1)} \rangle = B, \qquad \langle c^{(2)}, w^{(2)} \rangle = B. \tag{4}$$

We now compare the two optima using two scalings.

*Step 1: scale up $w^{(2)}$ to become feasible for $c^{(1)}$.* Because only seller $j$'s cost decreased, the cost of $w^{(2)}$ under $c^{(1)}$ is

$$\langle c^{(1)}, w^{(2)} \rangle = \langle c^{(2)}, w^{(2)} \rangle - (c_j^{(2)} - c_j^{(1)})w_j^{(2)} = B - \Delta w_j^{(2)}.$$

Define $\alpha := \frac{B}{B - \Delta w_j^{(2)}}$. Since $c_j^{(1)} > 0$ and $w_j^{(2)} \leq B/c_j^{(2)}$, we have $B - \Delta w_j^{(2)} \geq B \cdot \frac{c_j^{(1)}}{c_j^{(2)}} > 0$, hence $\alpha$ is well-defined and $\alpha \geq 1$. Let $\widetilde{w}^{(2)} := \alpha w^{(2)}$. Then $\widetilde{w}^{(2)} \in \mathcal{W}(c^{(1)})$ and, by degree-$(-1)$ homogeneity,

$$F(\widetilde{w}^{(2)}) = \alpha^{-1}F(w^{(2)}). \tag{5}$$

Optimality of $w^{(1)}$ under $c^{(1)}$ implies

$$F(w^{(1)}) \leq F(\widetilde{w}^{(2)}) = \alpha^{-1}F(w^{(2)}). \tag{6}$$

*Step 2: scale down $w^{(1)}$ to become feasible for $c^{(2)}$.* Under the higher cost $c^{(2)}$, the expenditure of $w^{(1)}$ is

$$\langle c^{(2)}, w^{(1)} \rangle = \langle c^{(1)}, w^{(1)} \rangle + \Delta w_j^{(1)} = B + \Delta w_j^{(1)}.$$

Define $\beta := \frac{B}{B+\Delta w_j^{(1)}} \in (0,1]$ and $\widehat{w}^{(1)} := \beta w^{(1)}$. Then $\widehat{w}^{(1)} \in \mathcal{W}(c^{(2)})$ and

$$F(\widehat{w}^{(1)}) = \beta^{-1} F(w^{(1)}). \tag{7}$$

Optimality of $w^{(2)}$ under $c^{(2)}$ implies

$$F(w^{(2)}) \le F(\widehat{w}^{(1)}) = \beta^{-1} F(w^{(1)}). \tag{8}$$

*Step 3: combine.* Multiplying (6) and (8) yields

$$F(w^{(1)}) F(w^{(2)}) \le \alpha^{-1} \beta^{-1} F(w^{(2)}) F(w^{(1)}).$$

Since $F(w^{(1)}), F(w^{(2)}) > 0$, we cancel to obtain $\alpha\beta \le 1$. Substituting the definitions of $\alpha, \beta$ gives

$$\frac{B}{B - \Delta w_j^{(2)}} \cdot \frac{B}{B + \Delta w_j^{(1)}} \le 1 \quad \Longleftrightarrow \quad (B - \Delta w_j^{(2)})(B + \Delta w_j^{(1)}) \ge B^2.$$

Expanding and canceling $B^2$ yields

$$\Delta B \left(w_j^{(1)} - w_j^{(2)}\right) - \Delta^2 w_j^{(1)} w_j^{(2)} \ge 0.$$

If $\Delta = 0$ then $c^{(1)} = c^{(2)}$ and the conclusion is trivial. If $\Delta > 0$, divide by $\Delta$ to get

$$B \left(w_j^{(1)} - w_j^{(2)}\right) \ge \Delta w_j^{(1)} w_j^{(2)} \ge 0,$$

which implies $w_j^{(1)} \ge w_j^{(2)}$, proving the desired inequality. $\qquad\square$

### D.2. Non-Monotonicity of Suboptimal Solutions

**Instance.** Let $k = 2$, $d = 2$, and

$$\Sigma_1 := \begin{pmatrix} 1 & 0 \\ 0 & 0.1 \end{pmatrix}, \qquad \Sigma_2 := \begin{pmatrix} 0.1 & 0 \\ 0 & 1 \end{pmatrix}.$$

Fix prices $p_2 = 1$ and $p_1 \in (0, \infty)$, and consider budget shares $\pi \in [0,1]$ with $\pi_1 = \pi$, $\pi_2 = 1 - \pi$ (so $\pi$ is a 1D simplex variable). Define the price-normalized information matrix

$$S_{p_1}(\pi) := \pi \frac{\Sigma_1}{p_1} + (1 - \pi)\Sigma_2, \tag{9}$$

and the A-optimal objective

$$U_{p_1}(\pi) := \mathrm{tr}\big(S_{p_1}(\pi)^{-1}\big). \tag{10}$$

Since $S_{p_1}(\pi)$ is diagonal for all $\pi$, the objective is explicit:

$$U_{p_1}(\pi) = \frac{1}{\pi/p_1 + 0.1(1 - \pi)} + \frac{1}{0.1\pi/p_1 + (1 - \pi)}. \tag{11}$$

The induced expected number of samples purchased from seller 1 under budget $B$ is

$$q_1(\pi; p_1) := \frac{B\pi}{p_1}. \tag{12}$$

Since $B$ is fixed throughout, monotonicity in procurement is equivalent to monotonicity of $\pi/p_1$.

**Lemma 5** (Optimal procurement is monotone on this instance). *Let $\pi^\star(p_1) \in \arg\min_{\pi \in [0,1]} U_{p_1}(\pi)$. Then:*

*(i) For $p_1 = 1$, the unique minimizer is $\pi^\star(1) = \frac{1}{2}$, hence $q_1(\pi^\star(1); 1) = \frac{B}{2}$.*

*(ii) For $p_1 = \frac{99}{100}$, the minimizer satisfies $\pi^\star(99/100) \approx 0.499364$ and*

$$\frac{\pi^\star(99/100)}{99/100} \approx 0.504408 \ > \ 0.5,$$

*so optimal procurement strictly increases when seller 1 becomes cheaper.*

*Proof.* (i) Plugging $p_1 = 1$ into (11) yields

$$U_1(\pi) = \frac{1}{0.1 + 0.9\pi} + \frac{1}{1 - 0.9\pi},$$

which is strictly convex on $(0, 1)$ and has derivative

$$U_1'(\pi) = -\frac{0.9}{(0.1 + 0.9\pi)^2} + \frac{0.9}{(1 - 0.9\pi)^2}.$$

Setting $U_1'(\pi) = 0$ gives $(0.1 + 0.9\pi)^2 = (1 - 0.9\pi)^2$, hence $0.1 + 0.9\pi = 1 - 0.9\pi$ and $\pi = \frac{1}{2}$. Strict convexity implies uniqueness.

(ii) For $p_1 = 99/100$, write $\alpha := 1/p_1 = 100/99$. Then (11) becomes

$$U_{p_1}(\pi) = \frac{1}{0.1 + \pi(\alpha - 0.1)} + \frac{1}{1 + \pi(0.1\alpha - 1)}.$$

This is strictly convex on $(0, 1)$ and its derivative is

$$U_{p_1}'(\pi) = -\frac{\alpha - 0.1}{(0.1 + \pi(\alpha - 0.1))^2} - \frac{0.1\alpha - 1}{(1 + \pi(0.1\alpha - 1))^2}.$$

Setting $U_{p_1}'(\pi) = 0$ yields

$$\frac{\alpha - 0.1}{(0.1 + \pi(\alpha - 0.1))^2} = \frac{1 - 0.1\alpha}{(1 + \pi(0.1\alpha - 1))^2}.$$

Taking square roots (all terms are positive at the minimizer) gives a linear equation in $\pi$, hence a unique closed form minimizer. Substituting $\alpha = 100/99$ yields $\pi^\star(99/100) \approx 0.4993641102$, hence $\pi^\star(99/100)/(99/100) \approx 0.5044081921 > 0.5$. □

Consider the following one-step projected gradient method for minimizing $U_{p_1}$:

$$\pi^{(0)} := \frac{1}{2}, \qquad \pi^{(1)} := \Pi_{[0,1]}\left(\pi^{(0)} - \eta\,\widehat{g}(p_1)\right), \tag{13}$$

where $\eta > 0$ is a step size and $\widehat{g}(p_1)$ is an approximate gradient satisfying

$$\left|\widehat{g}(p_1) - U_{p_1}'(\pi^{(0)})\right| \le \epsilon. \tag{14}$$

This scalar oracle is equivalent to Definition 1 in the $k = 2$ simplex, since the feasible direction is 1D (and $\ell_2$ error in the full gradient implies an $\ell_2$ bound on the derivative along the feasible direction).

**Example 1** (Non-monotonicity under a valid $\epsilon$-oracle). *Fix $\eta := 1/10$ and $\epsilon := 1/25$. There exist oracle outputs $\widehat{g}(1)$ and $\widehat{g}(99/100)$ satisfying (14) such that the induced procurement*

$$\widehat{q}_1(p_1) := \frac{B\,\pi^{(1)}(p_1)}{p_1}$$

*violates monotonicity:*

$$\widehat{q}_1(99/100) \ < \ \widehat{q}_1(1), \tag{15}$$

*even though seller 1 is cheaper at $p_1 = 99/100$.*

*Proof.* We compute the true derivatives at $\pi^{(0)} = 1/2$.

*Case $p_1 = 1$.* From the proof of Lemma 5, $U_1'(\frac{1}{2}) = 0$. Choose the (valid) oracle output $\widehat{g}(1) := -\epsilon = -\frac{1}{25}$. Then (13) yields (projection is inactive)

$$\pi^{(1)}(1) = \frac{1}{2} - \frac{1}{10}\left(-\frac{1}{25}\right) = \frac{63}{125} = 0.504, \qquad \widehat{q}_1(1) = B \cdot 0.504.$$

*Case $p_1 = 99/100$.* A direct calculation from (11) yields the exact derivative at $\pi^{(0)} = 1/2$:

$$U'_{99/100}\left(\frac{1}{2}\right) = \frac{176805684}{14349883681} \approx 0.01232105346.$$

Choose the (valid) oracle output

$$\widehat{g}(99/100) := U'_{99/100}(1/2) + \epsilon,$$

which satisfies (14) with equality. Then (13) yields

$$\pi^{(1)}(99/100) = \frac{1}{2} - \frac{1}{10}\left(U'_{99/100}(1/2) + \frac{1}{25}\right) \approx 0.49476789465.$$

Therefore

$$\widehat{q}_1(99/100) = \frac{B\,\pi^{(1)}(99/100)}{99/100} \approx B \cdot 0.49976555016.$$

Comparing with $\widehat{q}_1(1) = B \cdot 0.504$ gives (15). $\qquad\square$

**Remark 1** (Interpretation). *Example 1 shows that* limited compute *(modeled as a coarse $\epsilon$-approximate gradient oracle and early stopping) can break monotonicity even in an instance where the* exact *optimum is monotone (Lemma 5). This is exactly the incentive pathology: a non-monotone allocation rule cannot be coupled with truthful Myerson payments.*

### D.3. Restoring Monotonicity

**Lemma 6** (Uniform approximation implies $\kappa$-monotonicity). *Let $w^\star(\cdot)$ be a monotone allocation rule in the sense of Definition 2 with $\kappa = 0$. Let $\widehat{w}(\cdot)$ be any other rule such that for some $\delta \geq 0$,*

$$\sup_c \|\widehat{w}(c) - w^\star(c)\|_\infty \leq \delta. \tag{16}$$

*Then $\widehat{w}$ is $2\delta$-monotone: for every $j$, every fixed $c_{-j}$, and all $c_j < c_j'$,*

$$\widehat{w}_j(c_j', c_{-j}) \ \leq \ \widehat{w}_j(c_j, c_{-j}) + 2\delta.$$

*Proof.* Fix $j$, $c_{-j}$, and $c_j < c_j'$. By monotonicity of $w^\star$, $w_j^\star(c_j', c_{-j}) \leq w_j^\star(c_j, c_{-j})$. Using (16) twice,

$$\widehat{w}_j(c_j', c_{-j}) \leq w_j^\star(c_j', c_{-j}) + \delta \leq w_j^\star(c_j, c_{-j}) + \delta \leq \widehat{w}_j(c_j, c_{-j}) + 2\delta.$$

$\qquad\square$

**Strong convexification and the "free gradient" property.** Let $f_c(w)$ denote the buyer's acquisition objective for cost profile $c$. Under compute constraints, the buyer runs projected gradient descent with an approximate gradient oracle and stops after $T$ iterations. Left unmitigated, the output need not be monotone. To enforce monotonicity, we consider adding a strongly convex term:

$$f_c^{(\alpha)}(w) := f_c(w) + \alpha R(w),$$

where $R$ is 1-strongly convex on the feasible region, and $\alpha > 0$ is a tunable strength.

Crucially, the gradient of the regularizer $\nabla R(w)$ is a known analytic function that can be computed *exactly and for free*, without querying the costly data attribution oracle. Therefore, at iteration $t$, the costly oracle is only needed to estimate

$\nabla f_c(w^{(t)})$. If the oracle returns $\tau^{(t)}$ satisfying $\|\tau^{(t)} - \nabla f_c(w^{(t)})\|_2 \le \epsilon_t$ at compute cost $\gamma(\epsilon_t)$, the combined gradient estimate $\tau^{(t)} + \alpha\nabla R(w^{(t)})$ approximates the regularized gradient $\nabla f_c^{(\alpha)}(w^{(t)})$ with the *exact same absolute error* $\epsilon_t$, and *no additional compute cost*.

**Monotonicity of the regularized optimum.** Let $w_\alpha^\star(c) := \arg\min_{w \in \mathcal{W}(c)} f_c^{(\alpha)}(w)$ be the exact minimizer of the regularized objective. Because the regularizer $R(w)$ is independent of the private costs $c$, adding it preserves the single-crossing properties that govern the comparative statics of the minimizer. Consequently, strong convexification preserves the fundamental incentive structure of the problem.

**Proposition 7** (Last-iterate bound for inexact PGD on a strongly convex objective). *Let $K \subset \mathbb{R}^k$ be a closed convex set. Let $g : K \to \mathbb{R}$ be $\mu$-strongly convex and $L$-smooth w.r.t. $\|\cdot\|_2$ (with $\mu > 0$), and let $x^\star = \arg\min_{x \in K} g(x)$. Run projected gradient descent with inexact gradients:*

$$x^{(t+1)} := \Pi_K\left(x^{(t)} - \eta\tau^{(t)}\right), \qquad \|\tau^{(t)} - \nabla g(x^{(t)})\|_2 \le \varepsilon_t,$$

*with step size $\eta \in (0, 1/L]$. Then for $\rho := 1 - \mu\eta \in (0, 1)$,*

$$\|x^{(T)} - x^\star\|_2 \le \rho^T \|x^{(0)} - x^\star\|_2 + \eta \sum_{t=0}^{T-1} \rho^{T-1-t}\varepsilon_t. \tag{17}$$

*Proof.* Let $e_t := \tau^{(t)} - \nabla g(x^{(t)})$, so $\|e_t\|_2 \le \varepsilon_t$. Since $x^\star$ minimizes $g$ over $K$, it satisfies the fixed-point condition $x^\star = \Pi_K(x^\star - \eta\nabla g(x^\star))$. By nonexpansiveness of Euclidean projection,

$$\|x^{(t+1)} - x^\star\|_2 = \left\|\Pi_K\big(x^{(t)} - \eta(\nabla g(x^{(t)}) + e_t)\big) - \Pi_K\big(x^\star - \eta\nabla g(x^\star)\big)\right\|_2$$
$$\le \|x^{(t)} - x^\star - \eta(\nabla g(x^{(t)}) - \nabla g(x^\star))\|_2 + \eta\|e_t\|_2.$$

Because $g$ is $\mu$-strongly convex and $L$-smooth and $\eta \le 1/L$, the gradient map is a contraction: $\|x - \eta\nabla g(x) - (y - \eta\nabla g(y))\|_2 \le (1 - \mu\eta)\|x - y\|_2$ (a standard fact; e.g., via the mean value theorem and $\mu I \preceq \nabla^2 g \preceq LI$). Thus

$$\|x^{(t+1)} - x^\star\|_2 \le \rho\|x^{(t)} - x^\star\|_2 + \eta\varepsilon_t.$$

Unrolling this recursion gives (17). $\square$

**Theorem 3** (Strong convexification guarantees $\xi$-monotonicity for *any* compute budget). *Fix a class of cost profiles $\mathcal{C} \subset \mathbb{R}_{>0}^k$. Assume $c \mapsto w^\star(c)$ is monotone, and $f_c$ is convex and $L$-smooth. Let $R$ be 1-strongly convex. For a fixed compute budget $C$ and target monotonicity $\xi > 0$, run inexact PGD on $f_c^{(\alpha)}$ for $T \ge 1$ steps with step size $\eta = 1/(L + \alpha)$, and oracle accuracies $\{\epsilon_t\}_{t=0}^{T-1}$ satisfying $\sum_{t=0}^{T-1} \gamma(\epsilon_t) \le C$.*

*Let $\widehat{w}_\alpha(c)$ denote the output. Let $D := \sup_{c \in \mathcal{C}} \|w^{(0)}(c) - w_\alpha^\star(c)\|_2$ and $\rho := \frac{L}{L+\alpha}$. For every $c \in \mathcal{C}$, the optimization error is bounded by:*

$$\|\widehat{w}(c) - w_\alpha^\star(c)\|_\infty \le \rho^T D + \eta \sum_{t=0}^{T-1} \rho^{T-1-t}\epsilon_t. \tag{18}$$

*Consequently, if the RHS of (18) is at most $\delta$, then $\widehat{w}$ is $2\delta$-monotone by Lemma 6. Furthermore, as $\alpha \to \infty$, we have $\rho \to 0$ and $\eta \to 0$. Thus, for any finite compute budget $C$ and any horizon $T \ge 1$, there exists an $\alpha$ sufficiently large such that the algorithm is $\xi$-monotone.*

*Proof.* Fix $c \in \mathcal{C}$. Apply Proposition 7 with $g = f_c^{(\alpha)}$ and $K = \mathcal{W}(c)$. Note that $g$ is $\alpha$-strongly convex and $(L+\alpha)$-smooth. With $\eta = 1/(L + \alpha)$, the contraction factor is $\rho = 1 - \alpha\eta = L/(L + \alpha)$. By the standard $\ell_2$ bound:

$$\|\widehat{w}(c) - w_\alpha^\star(c)\|_2 \le \rho^T \|w^{(0)}(c) - w_\alpha^\star(c)\|_2 + \eta \sum_{t=0}^{T-1} \rho^{T-1-t}\epsilon_t.$$

Using $\|\cdot\|_\infty \le \|\cdot\|_2$ gives the desired inequality. Since $w_\alpha^\star(c)$ is exactly monotone, Lemma 6 yields $2\delta$-monotonicity. $\square$

**The Compute-Truthfulness-Utility Trilemma.** Theorem 3 reveals a fundamental trade-off. We can secure $\xi$-monotonicity (Truthfulness) for *any* compute budget (Compute) by making $\alpha$ large. This shrinks the condition number of the optimization problem, allowing rapid convergence. However, this truthfulness guarantee is paid for with *utility* (learning performance). A massive $\alpha$ introduces a *bias*, pulling the allocation away from the utility-maximizing $w^\star(c)$ and towards the data-agnostic minimizer of $R(w)$ (e.g., the uniform distribution).

We can bound this utility loss by assuming a quadratic-growth condition for $f_c$.

**Definition 4** (Quadratic growth). *We say $f_c$ satisfies $\zeta$-quadratic growth on $\mathcal{W}(c)$ if there exists $\zeta > 0$ such that for every $w \in \mathcal{W}(c)$,*

$$f_c(w) - \min_{u \in \mathcal{W}(c)} f_c(u) \;\geq\; \frac{\zeta}{2}\|w - w^\star(c)\|_2^2,$$

*where $w^\star(c) \in \arg\min_{\mathcal{W}(c)} f_c$ is any minimizer.*

**Lemma 7** (Regularization bias hurts utility). *Assume $f_c$ satisfies $\zeta$-quadratic growth on $\mathcal{W}(c)$. Assume $R$ is nonnegative on $\mathcal{W}(c)$ and let $R_{\max}(c) := \sup_{w \in \mathcal{W}(c)} R(w) < \infty$. Then the regularized minimizer $w_\alpha^\star(c)$ satisfies*

$$\|w_\alpha^\star(c) - w^\star(c)\|_2 \;\leq\; \sqrt{\frac{2\alpha R_{\max}(c)}{\zeta}}. \tag{19}$$

*Proof of Lemma 7.* Let $w^\star(c)$ be a minimizer of the original objective $f_c$, and let $w_\alpha^\star(c)$ be the minimizer of the regularized objective $f_c^{(\alpha)} = f_c + \alpha R$.

By the definition of $w_\alpha^\star(c)$ as the minimizer of $f_c^{(\alpha)}$, its regularized objective value must be less than or equal to the regularized objective value of any other point, including $w^\star(c)$:

$$f_c(w_\alpha^\star(c)) + \alpha R(w_\alpha^\star(c)) \;\leq\; f_c(w^\star(c)) + \alpha R(w^\star(c)).$$

Since $R(w) \leq R_{\max}(c)$ for all $w \in \mathcal{W}(c)$, we can upper bound the right-hand side:

$$f_c(w_\alpha^\star(c)) + \alpha R(w_\alpha^\star(c)) \;\leq\; f_c(w^\star(c)) + \alpha R_{\max}(c).$$

Since $R(w) \geq 0$ for all $w \in \mathcal{W}(c)$ and $\alpha > 0$, the term $\alpha R(w_\alpha^\star(c))$ is non-negative. Dropping this non-negative term from the left-hand side preserves the inequality:

$$f_c(w_\alpha^\star(c)) \;\leq\; f_c(w^\star(c)) + \alpha R_{\max}(c).$$

Rearranging this inequality bounds the sub-optimality of $w_\alpha^\star(c)$ with respect to the original objective function $f_c$:

$$f_c(w_\alpha^\star(c)) - f_c(w^\star(c)) \;\leq\; \alpha R_{\max}(c). \tag{20}$$

Finally, we apply the $\zeta$-quadratic growth condition of $f_c$ (Definition 4) evaluated at the point $w = w_\alpha^\star(c)$:

$$\frac{\zeta}{2}\|w_\alpha^\star(c) - w^\star(c)\|_2^2 \;\leq\; f_c(w_\alpha^\star(c)) - f_c(w^\star(c)).$$

Chaining this with inequality (20) yields:

$$\frac{\zeta}{2}\|w_\alpha^\star(c) - w^\star(c)\|_2^2 \;\leq\; \alpha R_{\max}(c).$$

Multiplying by $\frac{2}{\zeta}$ and taking the square root completes the proof:

$$\|w_\alpha^\star(c) - w^\star(c)\|_2 \;\leq\; \sqrt{\frac{2\alpha R_{\max}(c)}{\zeta}}.$$

$\square$

Thus, to maximize learning performance while maintaining approximate truthfulness, the system designer should choose the *minimum* $\alpha$ such that the RHS of (18) is $\leq \xi/2$.

**Choosing the optimal strongly convex component (minimax drift).**

Theorem 3 and Lemma 7 suggest two levers for the mechanism designer: (i) the curvature strength $\alpha$ (which buys truthfulness but hurts utility); (ii) the geometry of $R$ (which controls both optimization conditioning and the utility drift $R_{\max}$).

A clean notion of "not moving too far" from the original objective is the worst-case *additive drift* in objective values over the feasible set:

$$\sup_{w \in \mathcal{W}(c)} \left| f_c^{(\alpha)}(w) - f_c(w) \right| = \alpha \sup_{w \in \mathcal{W}(c)} R(w) \leq \alpha R_{\max}.$$

Thus, for a *fixed* strong convexity level $\alpha$ (chosen to enforce truthfulness), it is natural to choose $R$ that minimizes $R_{\max}$ subject to 1-strong convexity on the domain.

Moreover, in order to ensure that the $\xi$-monotonicity guarantee is preserved we can choose the smallest $\alpha$ so that

$$\rho^T D + \eta \sum_{t=0}^{T-1} \rho^{T-1-t} \epsilon_t \leq \frac{\xi}{2},$$

where $\rho = L/(L+\alpha)$.

### D.4. Implicit Payment Computation

**Procurement setting and notation.** We work in the single-parameter procurement model from Appendix A. There are $k$ sellers. Seller $i$ has a private per-unit cost $c_i \in [\underline{c}_i, \overline{c}_i] \subset \mathbb{R}_{>0}$ and reports $\hat{c}_i$. A (possibly randomized) *acquisition algorithm* maps reports to a nonnegative allocation $\boldsymbol{n}(\hat{\boldsymbol{c}}) \in \mathbb{R}_{\geq 0}^k$. We assume $A$ enforces *reported induced-cost feasibility*:

$$\sum_{i=1}^{k} \hat{c}_i \, n_i(\hat{\boldsymbol{c}}) \leq B \qquad \text{and} \qquad n_i(\hat{\boldsymbol{c}}) \geq 0 \ \ \forall i. \tag{21}$$

The mechanism pays seller $i$ an amount $p_i(\hat{\boldsymbol{c}})$, yielding quasi-linear utility $u_i(\hat{\boldsymbol{c}}; c_i) = p_i(\hat{\boldsymbol{c}}) - c_i \, n_i(\hat{\boldsymbol{c}})$.

**Monotonicity.** Fix $i$ and fix $\hat{\boldsymbol{c}}_{-i}$. A (deterministic) allocation rule is *monotone (procurement form)* if the map $\hat{c}_i \mapsto n_i(\hat{c}_i, \hat{\boldsymbol{c}}_{-i})$ is weakly *non-increasing*. If $A$ is randomized, we define the *interim* allocation $\bar{n}_i(\hat{c}_i, \hat{\boldsymbol{c}}_{-i}) := \mathbb{E}[n_i(\hat{c}_i, \hat{\boldsymbol{c}}_{-i})]$, where the expectation is over the internal random seed of $A$ (and/or oracle randomness). We say $A$ is *monotone-in-expectation* if $\bar{n}_i(\cdot, \hat{\boldsymbol{c}}_{-i})$ is weakly non-increasing.

**Myerson's payment identity (procurement form).** For completeness we restate and prove the standard identity used in Appendix A.

**Lemma 8** (Myerson payments for procurement). *Fix seller $i$ and $\hat{\boldsymbol{c}}_{-i}$. Let $\bar{n}_i(\cdot) := \bar{n}_i(\cdot, \hat{\boldsymbol{c}}_{-i})$ be nonnegative and weakly non-increasing on $[\underline{c}_i, \overline{c}_i]$. Define the* interim payment

$$\bar{p}_i(\hat{c}_i, \hat{\boldsymbol{c}}_{-i}) := \hat{c}_i \, \bar{n}_i(\hat{c}_i) + \int_{\hat{c}_i}^{\overline{c}_i} \bar{n}_i(t) \, dt. \tag{22}$$

*Then truthful reporting maximizes seller $i$'s expected utility (dominant strategy in expectation), and moreover the truthful expected utility satisfies $\bar{u}_i(c_i, \hat{\boldsymbol{c}}_{-i}; c_i) = \int_{c_i}^{\overline{c}_i} \bar{n}_i(t) \, dt \geq 0$.*

*Proof.* Fix true cost $c_i$ and consider an arbitrary report $r \in [\underline{c}_i, \overline{c}_i]$. Using (22), seller $i$'s expected utility under report $r$ equals

$$U(r) := \bar{p}_i(r, \hat{\boldsymbol{c}}_{-i}) - c_i \, \bar{n}_i(r, \hat{\boldsymbol{c}}_{-i})$$

$$= (r - c_i) \, \bar{n}_i(r) + \int_r^{\overline{c}_i} \bar{n}_i(t) \, dt.$$

We show $U(r) \leq U(c_i)$ for all $r$.

If $r > c_i$, then

$$U(r) - U(c_i) = (r - c_i)\,\bar{n}_i(r) - \int_{c_i}^{r} \bar{n}_i(t)\,dt.$$

Since $\bar{n}_i$ is non-increasing, $\bar{n}_i(t) \geq \bar{n}_i(r)$ for all $t \in [c_i, r]$, hence $\int_{c_i}^{r} \bar{n}_i(t)\,dt \geq (r - c_i)\bar{n}_i(r)$ and therefore $U(r) - U(c_i) \leq 0$.

If $r < c_i$, then

$$U(r) - U(c_i) = (r - c_i)\,\bar{n}_i(r) + \int_{r}^{c_i} \bar{n}_i(t)\,dt.$$

Now $\bar{n}_i(t) \leq \bar{n}_i(r)$ for all $t \in [r, c_i]$, hence $\int_{r}^{c_i} \bar{n}_i(t)\,dt \leq (c_i - r)\bar{n}_i(r)$, implying $U(r) - U(c_i) \leq (r - c_i)\bar{n}_i(r) + (c_i - r)\bar{n}_i(r) = 0$.

Thus $r = c_i$ maximizes expected utility. Finally, substituting $r = c_i$ gives $U(c_i) = \int_{c_i}^{\bar{c}_i} \bar{n}_i(t)\,dt \geq 0$, proving expected IR under truth-telling. $\square$

**Why** (22) **is computationally expensive.**     Computing (22) requires evaluating $t \mapsto \bar{n}_i(t, \hat{\boldsymbol{c}}_{-i})$ over a continuum of $t$. If $A$ is an oracle-driven acquisition pipeline, each evaluation of $n_i(t, \hat{\boldsymbol{c}}_{-i})$ entails *additional attribution-oracle compute*. The goal of this subsection is to eliminate those extra calls.

**Roadmap.**    We introduce (i) a one-sample identity that expresses an integral as an expectation, and (ii) an *upward self-resampling* procedure for costs that ensures the expectation we compute is taken over the *correct* (fixed-point) allocation rule. Our approach is an adaptation of Babaioff et al. (2015).

**Step 1: A one-sample identity for integrals.**

**Lemma 9** (One-sample integral identity).  *Let $I = (a, b)$ be an open interval and let $F : I \to (0, 1)$ be strictly increasing and continuously differentiable, with $\lim_{z \downarrow a} F(z) = 0$ and $\lim_{z \uparrow b} F(z) = 1$. Let $Y$ be a random variable on $I$ with CDF $F$ and density $f = F'$. Then for every measurable $g : I \to \mathbb{R}$ with $\int_I |g(z)|\,dz < \infty$,*

$$\int_I g(z)\,dz \;=\; \mathbb{E}\!\left[\frac{g(Y)}{f(Y)}\right].$$

*Proof.* Since $Y$ has density $f$, for any measurable $h$ with $\int_I |h(z)|f(z)\,dz < \infty$ we have $\mathbb{E}[h(Y)] = \int_I h(z)f(z)\,dz$. Taking $h(z) = g(z)/f(z)$ yields $\mathbb{E}[g(Y)/f(Y)] = \int_I g(z)\,dz$. $\square$

**Step 2: Upward self-resampling for costs.**     To apply Lemma 9 to the integral term in (22) *without* extra runs of $A$, we randomize the reported costs *upwards* (to worse types) before calling the allocation algorithm, and we use the realized allocation to build an unbiased estimator.

The key difficulty is that resampling changes the allocation rule whose integral we must estimate. Following Babaioff et al. (2015), we define a self-resampling procedure with a "fixed-point" property ensuring that conditioning on the sampled point $Y_i = t$ makes the observed allocation distributed exactly as if seller $i$ had reported $t$.

**Definition 5** (Upward self-resampling procedure).  *Fix seller $i$ and fix $\alpha \in (0, 1)$. An* upward self-resampling procedure *is a randomized map that takes as input $\hat{c}_i \in [\underline{c}_i, \bar{c}_i]$ and outputs a pair of random variables $(X_i, Y_i) \in [\underline{c}_i, \bar{c}_i]^2$ such that:*

(a) *(**Monotonicity in input**) For every fixed random seed, the maps $\hat{c}_i \mapsto X_i(\hat{c}_i)$ and $\hat{c}_i \mapsto Y_i(\hat{c}_i)$ are weakly non-decreasing.*

(b) *(**No-change w.p.** $1 - \alpha$) With probability $1 - \alpha$, $X_i(\hat{c}_i) = Y_i(\hat{c}_i) = \hat{c}_i$. Otherwise, $\hat{c}_i < Y_i(\hat{c}_i) \leq X_i(\hat{c}_i) \leq \bar{c}_i$.*

(c) *(**Fixed-point / self-similarity**) For any $t \in (\hat{c}_i, \bar{c}_i)$, the conditional distribution of $X_i(\hat{c}_i)$ given $\{Y_i(\hat{c}_i) = t\}$ equals the unconditional distribution of $X_i(t)$:*

$$\Pr[X_i(\hat{c}_i) \leq a \mid Y_i(\hat{c}_i) = t] \;=\; \Pr[X_i(t) \leq a] \qquad \forall a \in [\underline{c}_i, \bar{c}_i].$$

*(d) (**Smooth conditional law of** $Y_i$) Conditional on the event $\{Y_i(\hat{c}_i) > \hat{c}_i\}$, the random variable $Y_i(\hat{c}_i)$ has a continuously differentiable CDF on $(\hat{c}_i, \overline{c}_i)$ with everywhere-positive density $f_i(\cdot; \hat{c}_i)$.*

**A canonical upward procedure (bounded interval).** We now give an explicit construction on $[\underline{c}_i, \overline{c}_i]$ which is the procurement analogue of the canonical self-resampling procedure in Babaioff et al. (2015). It will be convenient to define a helper random variable $U \sim \mathrm{Unif}[0, 1]$.

**Definition 6** (Canonical upward self-resampling). *Fix $\alpha \in (0, 1)$ and $\hat{c}_i \in [\underline{c}_i, \overline{c}_i]$. Define $(X_i, Y_i)$ by:*

*(i) If $\hat{c}_i = \overline{c}_i$, set $X_i = Y_i = \overline{c}_i$ deterministically.*

*(ii) Otherwise, with probability $1 - \alpha$ set $X_i = Y_i = \hat{c}_i$.*

*(iii) With the remaining probability $\alpha$: sample $Y_i := \hat{c}_i + (\overline{c}_i - \hat{c}_i)U$ (uniform on $[\hat{c}_i, \overline{c}_i]$), and then define $X_i$ by the recursion*

$$X_i := \mathrm{Rec}(Y_i),$$

*where $\mathrm{Rec}(t)$ is defined as: with probability $1 - \alpha$ return $t$, else sample $t' := t + (\overline{c}_i - t)U'$ (fresh $U' \sim \mathrm{Unif}[0, 1]$) and return $\mathrm{Rec}(t')$.*

**Lemma 10** (Canonical upward resampling satisfies Definition 5). *The procedure in Definition 6 is an upward self-resampling procedure. Moreover, for every $\hat{c}_i < \overline{c}_i$, conditional on $\{Y_i(\hat{c}_i) > \hat{c}_i\}$ we have $Y_i(\hat{c}_i) \sim \mathrm{Unif}[\hat{c}_i, \overline{c}_i]$, so the conditional density is constant:*

$$f_i(t; \hat{c}_i) = \frac{1}{\overline{c}_i - \hat{c}_i}, \qquad t \in (\hat{c}_i, \overline{c}_i). \tag{23}$$

*Proof.* We verify the four properties in Definition 5.

*(a) Monotonicity in input.* Fix the entire random seed (an infinite sequence of i.i.d. uniforms used by the procedure). Each time the procedure samples a point uniformly in an interval $[s, \overline{c}_i]$ it returns $s + (\overline{c}_i - s)U = (1 - U)s + U\overline{c}_i$, which is weakly increasing in $s$ for fixed $U$. Because the recursion composes such maps, both outputs $Y_i(\hat{c}_i)$ and $X_i(\hat{c}_i)$ are weakly increasing functions of $\hat{c}_i$ for every fixed seed.

*(b) No-change with probability $1 - \alpha$.* This holds by construction (steps (i)–(iii)).

*(c) Fixed-point / self-similarity.* Fix $\hat{c}_i < \overline{c}_i$ and fix $t \in (\hat{c}_i, \overline{c}_i)$. On the event $\{Y_i(\hat{c}_i) = t\}$ the algorithm (by definition) proceeds to generate $X_i$ by running the recursive procedure $\mathrm{Rec}$ starting from input $t$, using fresh randomness that is independent of the randomness that selected $Y_i$. But the unconditional definition of $X_i(t)$ is *exactly* the output of the same recursive procedure $\mathrm{Rec}$ started from input $t$ with fresh randomness. Therefore, the conditional law of $X_i(\hat{c}_i)$ given $\{Y_i(\hat{c}_i) = t\}$ equals the unconditional law of $X_i(t)$, as required.

*(d) Smooth conditional law of $Y_i$.* Conditional on the resampling branch (which occurs with probability $\alpha$), the random variable $Y_i$ is uniform on $[\hat{c}_i, \overline{c}_i]$ by construction. This implies that the conditional CDF is differentiable and strictly increasing on $(\hat{c}_i, \overline{c}_i)$ with density given in (23). $\qquad\square$

**Step 3: The zero-overhead mechanism (one call to $A$).** Fix $\alpha \in (0, 1)$ and assume we have (independently across $i$) an upward self-resampling procedure $(X_i, Y_i)$ satisfying Definition 5 (e.g., the canonical procedure above). Given reported costs $\hat{\boldsymbol{c}}$, the mechanism proceeds:

**Step 1:** Independently for each seller $i$, draw $(X_i, Y_i)$ from the upward self-resampling procedure on input $\hat{c}_i$. Let $\boldsymbol{X} = (X_1, \ldots, X_k)$ and $\boldsymbol{Y} = (Y_1, \ldots, Y_k)$.

**Step 2:** **(Single call)** Run the acquisition algorithm *once* on $\boldsymbol{X}$ and output $\boldsymbol{n} := A(\boldsymbol{X})$.

**Step 3:** Pay each seller $i$ the random payment

$$\tilde{p}_i(\hat{\boldsymbol{c}}) := \hat{c}_i \, n_i + \tilde{R}_i, \qquad \tilde{R}_i := \begin{cases} \dfrac{1}{\alpha} \cdot \dfrac{n_i}{f_i(Y_i; \hat{c}_i)} & \text{if } Y_i > \hat{c}_i, \\[2mm] 0 & \text{if } Y_i = \hat{c}_i. \end{cases} \tag{24}$$

In particular, under the canonical upward procedure (Lemma 10), this simplifies to

$$\tilde{p}_i(\hat{\boldsymbol{c}}) = \hat{c}_i \, n_i \;+\; \mathbf{1}\{Y_i > \hat{c}_i\} \cdot \frac{\overline{c}_i - \hat{c}_i}{\alpha} \, n_i. \tag{25}$$

**Correctness guarantee.** We now prove that (24) implements the Myerson payment identity (22) in expectation, and therefore yields a truthful-in-expectation mechanism.

**Theorem 4** (Zero-overhead implicit payments for procurement). *Assume:*

(a) *(**Ex-post monotonicity**) For each seller $i$, for every fixed realization of all randomness in $A$ (and any "nature" randomness, e.g., stochastic oracles), and for every fixed $\boldsymbol{x}_{-i}$, the map $x_i \mapsto n_i(\boldsymbol{x}_{-i}, x_i)$ is weakly non-increasing.*

(b) *(**Independence**) The resampling randomness $\{(X_i, Y_i)\}_{i=1}^{k}$ is mutually independent and independent of all randomness used inside $A$.*

*Then the mechanism defined by (24) satisfies:*

(i) ***Single-call / zero-overhead:** it makes exactly one call to $A$.*

(ii) ***Truthfulness-in-expectation (ex-post truthful w.r.t. nature):** for every seller $i$, for every fixed realization of nature randomness inside $A$, and for every fixed $\hat{\boldsymbol{c}}_{-i}$, truthful reporting maximizes expected utility over the mechanism's random seed.*

(iii) ***Universal ex-post IR:** for every realization of the mechanism's random seed, if seller $i$ reports truthfully then her realized utility is nonnegative.*

(iv) ***Allocation preservation with high probability:** with probability at least $1 - k\alpha$ we have $\boldsymbol{X} = \hat{\boldsymbol{c}}$ coordinatewise, so the realized allocation equals $A(\hat{\boldsymbol{c}})$.*

*Proof.* Fix seller $i$ and fix $\hat{\boldsymbol{c}}_{-i}$. Let $Q$ denote the combined random seed consisting of: (i) all resampling randomness for all sellers, and (ii) the internal randomness of $A$ and nature. Write $\boldsymbol{n}(\hat{c}_i; Q) := A(\boldsymbol{X}(\hat{c}_i, \hat{\boldsymbol{c}}_{-i}); Q)$ for the realized allocation.

*Step 1: Define the interim allocation rule induced by resampling.* Define

$$\bar{n}_i(\hat{c}_i, \hat{\boldsymbol{c}}_{-i}) := \mathbb{E}\big[n_i(\hat{c}_i; Q)\big],$$

where the expectation is over the mechanism's random seed (the resampling) and over any random seed of $A$, holding fixed the reports.

*Step 2: Interim monotonicity.* Fix $\hat{c}_i < \hat{c}_i'$. By Definition 5(a), for every fixed resampling seed we have $X_i(\hat{c}_i) \leq X_i(\hat{c}_i')$. Holding $\boldsymbol{X}_{-i}$ fixed and fixing the internal random seed of $A$, assumption (a) implies $n_i(\boldsymbol{X}_{-i}, X_i(\hat{c}_i)) \geq n_i(\boldsymbol{X}_{-i}, X_i(\hat{c}_i'))$. Taking expectations over all random seeds yields $\bar{n}_i(\hat{c}_i, \hat{\boldsymbol{c}}_{-i}) \geq \bar{n}_i(\hat{c}_i', \hat{\boldsymbol{c}}_{-i})$, so $\bar{n}_i(\cdot, \hat{\boldsymbol{c}}_{-i})$ is weakly non-increasing.

*Step 3: Identify the expected payment with Myerson's identity.* We show that the expected payment satisfies

$$\mathbb{E}\big[\tilde{p}_i(\hat{c}_i, \hat{\boldsymbol{c}}_{-i})\big] = \hat{c}_i \, \bar{n}_i(\hat{c}_i, \hat{\boldsymbol{c}}_{-i}) + \int_{\hat{c}_i}^{\overline{c}_i} \bar{n}_i(t, \hat{\boldsymbol{c}}_{-i}) \, dt. \tag{26}$$

Given (26) and interim monotonicity, Lemma 8 implies truthfulness-in-expectation.

By (24), $\tilde{p}_i = \hat{c}_i n_i + \tilde{R}_i$, hence

$$\mathbb{E}[\tilde{p}_i] = \hat{c}_i \, \bar{n}_i(\hat{c}_i, \hat{\boldsymbol{c}}_{-i}) + \mathbb{E}[\tilde{R}_i].$$

Thus it suffices to show

$$\mathbb{E}[\tilde{R}_i] = \int_{\hat{c}_i}^{\overline{c}_i} \bar{n}_i(t, \hat{\boldsymbol{c}}_{-i}) \, dt. \tag{27}$$

Let $E$ be the event $\{Y_i > \hat{c}_i\}$. By Definition 5(b), $\Pr[E] = \alpha$ and $\tilde{R}_i = 0$ on $E^c$. Therefore

$$\mathbb{E}[\tilde{R}_i] = \Pr[E] \cdot \mathbb{E}[\tilde{R}_i \mid E] = \alpha \cdot \mathbb{E}\left[\frac{1}{\alpha} \cdot \frac{n_i}{f_i(Y_i; \hat{c}_i)} \,\bigg|\, E\right] = \mathbb{E}\left[\frac{n_i}{f_i(Y_i; \hat{c}_i)} \,\bigg|\, E\right]. \tag{28}$$

Next we relate the conditional expectation of $n_i$ given $Y_i$ to the interim allocation rule. Fix $t \in (\hat{c}_i, \overline{c}_i)$. By Definition 5(c) (self-similarity) and independence in assumption (b), the conditional distribution of the full resampled vector $\boldsymbol{X}$ given $\{Y_i = t\}$ is identical to the unconditional distribution of the vector that would be produced by running the same resampling procedures on the report profile $(t, \hat{\boldsymbol{c}}_{-i})$. Consequently,

$$\mathbb{E}[n_i \mid Y_i = t, E] = \bar{n}_i(t, \hat{\boldsymbol{c}}_{-i}). \tag{29}$$

(Here $E$ is redundant given $Y_i = t > \hat{c}_i$, but we keep it for clarity.)

Under the conditional law given $E$, the random variable $Y_i$ has density $f_i(\cdot; \hat{c}_i)$ on $(\hat{c}_i, \overline{c}_i)$ by Definition 5(d). Applying the law of iterated expectation and (29) to (28) yields

$$\mathbb{E}[\tilde{R}_i] = \mathbb{E}\left[\frac{\mathbb{E}[n_i \mid Y_i]}{f_i(Y_i; \hat{c}_i)}\right] = \mathbb{E}\left[\frac{\bar{n}_i(Y_i, \hat{\boldsymbol{c}}_{-i})}{f_i(Y_i; \hat{c}_i)}\right].$$

Now apply Lemma 9 on the interval $I = (\hat{c}_i, \overline{c}_i)$ with $g(t) = \bar{n}_i(t, \hat{\boldsymbol{c}}_{-i})$ and the CDF corresponding to $Y_i \mid E$. This gives exactly (27), which implies (26).

*Step 4: Universal ex-post IR.* Fix any realization of the mechanism random seed. Under truthful reporting $\hat{c}_i = c_i$, the realized payment is $\tilde{p}_i = c_i n_i + \tilde{R}_i$ with $\tilde{R}_i \geq 0$ always. Hence the realized utility is $\tilde{p}_i - c_i n_i = \tilde{R}_i \geq 0$, establishing universal ex-post IR.

*Step 5: Single-call and allocation preservation.* The procedure calls $A$ exactly once by construction. Moreover, for each $i$, Definition 5(b) implies $\Pr[X_i = \hat{c}_i] = 1 - \alpha$. Independence across sellers yields $\Pr[\forall i, X_i = \hat{c}_i] = (1-\alpha)^k \geq 1 - k\alpha$ (union bound), and on this event $\boldsymbol{X} = \hat{\boldsymbol{c}}$, so the realized allocation equals $A(\hat{\boldsymbol{c}})$. $\square$

**Feasibility under true costs.** The resampling is *upward* ($X_i \geq \hat{c}_i$). Thus if the algorithm enforces feasibility $\sum_i X_i n_i \leq B$ on the single call, then under truthful reporting $c_i = \hat{c}_i \leq X_i$ we have $\sum_i c_i n_i \leq \sum_i X_i n_i \leq B$, so the induced-cost cap is respected under truth-telling.

**Approximately monotone acquisition rules.** Theorem 4 assumes (exact) monotonicity. In compute-constrained acquisition, monotonicity may only hold approximately (e.g., due to early stopping, noisy gradients, or approximate projections). We now quantify how approximate monotonicity translates into approximate truthfulness, while still using the *same* implicit payment computation.

**Definition 7** ($\xi$-monotonicity (additive, interim, procurement))**.** *Fix seller $i$ and $\hat{\boldsymbol{c}}_{-i}$. The interim allocation rule $\bar{n}_i(\cdot, \hat{\boldsymbol{c}}_{-i})$ is $\xi$-monotone if for all $\hat{c}_i < \hat{c}_i'$,*

$$\bar{n}_i(\hat{c}_i', \hat{\boldsymbol{c}}_{-i}) \leq \bar{n}_i(\hat{c}_i, \hat{\boldsymbol{c}}_{-i}) + \xi.$$

**Theorem 5** ($\xi$-monotonicity implies $\epsilon$-DSIC-in-expectation)**.** *Fix seller $i$ and $\hat{\boldsymbol{c}}_{-i}$, and let $\bar{n}_i(\cdot) := \bar{n}_i(\cdot, \hat{\boldsymbol{c}}_{-i})$ be any nonnegative interim allocation rule on $[\underline{c}_i, \overline{c}_i]$. Define interim payments by the Myerson formula (22). If $\bar{n}_i(\cdot)$ is $\xi$-monotone, then truthful reporting is an $\epsilon$-dominant strategy in expectation, with*

$$\epsilon \leq \xi (\overline{c}_i - \underline{c}_i).$$

*That is, for every true $c_i$ and every report $r$,*

$$\bar{u}_i(r, \hat{\boldsymbol{c}}_{-i}; c_i) \leq \bar{u}_i(c_i, \hat{\boldsymbol{c}}_{-i}; c_i) + \xi(\overline{c}_i - \underline{c}_i).$$

*Proof.* Fix true cost $c_i$ and define $U(r)$ as in the proof of Lemma 8:

$$U(r) = (r - c_i)\bar{n}_i(r) + \int_r^{\overline{c}_i} \bar{n}_i(t)\, dt.$$

We bound $U(r) - U(c_i)$.

*Case 1:* $r > c_i$. Then

$$U(r) - U(c_i) = (r - c_i)\bar{n}_i(r) - \int_{c_i}^{r} \bar{n}_i(t)\, dt.$$

By $\xi$-monotonicity, for all $t \in [c_i, r]$ we have $\bar{n}_i(t) \geq \bar{n}_i(r) - \xi$ (rearranging the definition), hence

$$\int_{c_i}^{r} \bar{n}_i(t)\, dt \geq (r - c_i)(\bar{n}_i(r) - \xi).$$

Substituting gives

$$U(r) - U(c_i) \leq (r - c_i)\bar{n}_i(r) - (r - c_i)(\bar{n}_i(r) - \xi) = \xi(r - c_i).$$

*Case 2:* $r < c_i$. Then

$$U(r) - U(c_i) = (r - c_i)\bar{n}_i(r) + \int_{r}^{c_i} \bar{n}_i(t)\, dt.$$

By $\xi$-monotonicity, for all $t \in [r, c_i]$ we have $\bar{n}_i(t) \leq \bar{n}_i(r) + \xi$, hence

$$\int_{r}^{c_i} \bar{n}_i(t)\, dt \leq (c_i - r)(\bar{n}_i(r) + \xi).$$

Substituting yields

$$U(r) - U(c_i) \leq (r - c_i)\bar{n}_i(r) + (c_i - r)(\bar{n}_i(r) + \xi) = \xi(c_i - r).$$

Combining both cases, $U(r) - U(c_i) \leq \xi|r - c_i| \leq \xi(\bar{c}_i - \underline{c}_i)$. $\qquad\square$

**Corollary 2** (Zero-overhead $\epsilon$-DSIC under approximate monotonicity). *Suppose the conditions of Theorem 4 hold except that interim monotonicity is replaced by $\xi$-monotonicity (Definition 7). Then the single-call mechanism (24) is $\epsilon$-DSIC-in-expectation with $\epsilon \leq \xi(\bar{c}_i - \underline{c}_i)$ for seller $i$.*

*Proof.* The proof of Theorem 4 shows that the mechanism's expected payments satisfy the Myerson identity (22) for the interim allocation rule $\bar{n}_i$ (without using monotonicity). Theorem 5 then converts $\xi$-monotonicity into $\epsilon$-DSIC. $\qquad\square$

**Takeaway for compute-constrained data pricing.**     Once we have (exact or approximate) monotonicity of the acquisition rule (as discussed in Sections 3.3.1–3.3.3), we can obtain a (respectively exact or approximate) incentive-compatible mechanism *without additional calls* to the acquisition algorithm and therefore without additional attribution-oracle compute. This resolves the payment-computation overhead in compute-constrained settings.

