# OpenReview forum: "A Theory of Data Acquisition and Pricing at Scale"
_ICML.cc/2026/Conference — ICML 2026 regular_

### Official Review · Reviewer_NpZg · 2026-03-09

**Soundness:** 2
**Presentation:** 2
**Significance:** 2
**Originality:** 3
**Overall Recommendation:** 3
**Confidence:** 4

**Summary:**

The paper presents the data pricing under the compute constraints. The works is motivated by the setting where the data buyers can not make the data acquisition decisions optimally due to limited compute. The authors formalize the model as a 'pricing with an attribution oracle', where the buyer optimizes a data mixture vector using gradient information from a heuristic oracle, and adds mechanism design for truthful cost reporting.

**Compliance With Llm Reviewing Policy:**

Affirmed.

**Ethical Review Concerns:**

no concerns

**Final Justification:**

Since the authors did not respond, my concerns remain unresolved. The paper still relies on strong modeling assumptions that are central to its formulation, in particular that the buyer knows each seller’s distribution $D_i$ and, in the linear-regression instantiation, effectively has access to precomputed second moments $\sum_i$; this substantially limits practical realism.

**Key Questions For Authors:**

1. The authors assume each seller distribution $D_i$ to be known because it doesn't scale with oracle queries. In realistic procurement, estimating $\Sigma_i$ can be require accessing the data. How would the model and compute accounting change if $\Sigma_i$ is only available via samples?
2. The oracle model bounds approximation error in gradient estimates. Many practical attribution proxies can be systematically biased. Do your convergence results extend under biased gradient oracles, or do they critically require an unbiased error guarantee?
3. The authors restore approximate monotonicity by strong convexification, but this can force allocations towrd the regularizer. Can the authors quantify the utility loss induced by convexification and provide a principled way to select
$\alpha$.
4. The feasibility constraint is phrased in terms of induced costs. Many settings require post budget feasibility on payments. Why is expectation level feasibility the right model here?

**Limitations:**

yes

**Strengths And Weaknesses:**

Strengths:
1. Timely and well-motivated problem framing: The compute constraints are a real barrier for data pricing. Modeling them as an oracle-fidelity cost is a clean abstraction.
2. Clear theoretical landscape in a tractable setting: The linear-regression instantiation yields concrete insights, for instance., when selection is not worth compute; why fidelity should be back-loaded.
3. Mechanism-design angle is nontrivial: The monotonicity failure under early noisy gradients is an important point, and the single-call payment construction is potentially impactful if it generalizes.

Weakness:
1. Model assumptions are strong and drift from practice:
   a. The buyer is assumed to see each seller distribution $D_i$  and (in the linear model) effectively have access to moments as pre-computed. In real pipelines, estimating these moments itself can be expensive and noisy and may require access to the data you are trying to decide whether to buy.
  b. The oracle assumption with a know cost function is clean and but arguably too idealized for modern deep learning, where proxies can be biased and their error is hard to calibrate.
2. The procurement model uses Poisson sampling to enforce the budget in expectation and constrains true induced cost rather than the buyer's payments. This is defensible in some settings, but many data markets are payment budget constrained. The separation should be discussed more explicitly and positioned relative to standard budget feasible mechanism design.
3. While the theoretical development is substantial, the paper lacks empirical validation. Since the abstract states 'empirical analysis', this reads as a mismatch. Consider rewording this claim to avoid confusion.

---

> ### Author Rebuttal · Authors · 2026-03-31
>
> We would like to thank the reviewer for their constructive comments. We respond to the points they raised below:
>
> >Model assumptions and cost of learning the distributions: We agree that in some pipelines the distribution might need to be learned from data. Notice that for many downstream applications one can compute information about the data distribution without having to pay for the data; for instance, one might have free access to the data and can estimate several quantities about their distribution, but may have to pay for the right to use them in training pipelines. On other occasions, data providers might offer some small part of the dataset as a free sample so that buyers can get some information about its usefulness. That said, estimating quantities of interest of the distribution might incur computational cost. For instance, in our setting one could assume access to an oracle that takes as input samples and outputs a distribution that is “close” to the true one, under some appropriate notion of distance. Then, we would have to trade off the computational cost of estimating the distribution more accurately against the utility cost of running our algorithm on a misestimated distribution. While this would make the pipeline more realistic, it would add too many bells and whistles, and perhaps hinder the conceptual message of our work. We will elaborate on it in the next version of our work.
>
> >Biased oracles: Thanks for bringing this up, and we are happy to clarify it.  Our results do not rely on unbiasedness of the oracles. The assumption in the formal model is a bounded approximation error (or randomized high-probability analogue): the oracle output must be within $\varepsilon$ of the true gradient when we pay some cost $C(\varepsilon)$. Therefore, systematic bias is already allowed so long as we have control over it. What lies outside our present scope is arbitrary uncalibrated bias with no deterministic/high-probability control. In this regime it would be hopeless to make any theoretical claims. We will make this point more explicit in the revision.
>
> >Strongly convex surrogate: This is also a very good point. We have taken a first step towards addressing this question in Appendix D.3. There, we bound the optimization error of the regularized solution and the utility loss induced by regularization, and we also give a principled way to choose the smallest regularizer that achieves a desired monotonicity level. We will expand on this discussion and move it forward into the main text.
>
> >Feasibility constraint and Poisson sampling: This is an excellent point. First, regarding the Poisson sampling and the fact that budgets are forced in expectation, we view that as a mild restriction. Indeed, the concentration bounds we have used in our work show that this sampling scheme concentrates tightly around its expectation. Moreover, asking for constraints to hold in expectation is relatively standard in the mechanism design literature (e.g., BIC, budget-balancedness in expectation etc.). Appendix A.5 explains some applications where it is more natural to ask for cost feasibility rather than payment-budget feasibility. For instance, when buying data for specialized tasks (e.g., research-level mathematics / computer science), the bottleneck is usually not the budget of the buyer, but rather the effort / cost the data provider can put into the curation of the dataset. We do agree that there are applications where it is more natural to have budget constraints on the payments. Looking at the implicit payment computation scheme we develop (which is inspired by prior work), we can provide bounds on the difference between the total cost and the total payment. We will move those clarifications into the main text, and compare with classical budget-feasible literature in mechanism design.
>
> >Thanks for pointing out the empirical analysis part of the abstract. We will correct that in the next version of our work.

---

> > ### Author Rebuttal · Reviewer_NpZg · 2026-04-06
> >
> > The rebuttal addresses some points well, especially by clarifying that the theory only requires bounded oracle error rather than unbiasedness, and by pointing to Appendix D.3 for a bound on the utility loss from strong convexification and a principled choice of $\alpha$. However, the main concerns remain only partially addressed: the paper still assumes the buyer knows each seller’s distribution $D_i$ and effectively has access to precomputed $\sum_i$, which limits practical applicability. The justification for Poisson sampling and expectation-level feasibility is reasonable, but it does not fully resolve settings where hard payment-budget feasibility is the natural requirement.
> >
> > After careful consideration, I maintain my score.

---

### Official Review · Reviewer_db6A · 2026-03-11

**Soundness:** 3
**Presentation:** 2
**Significance:** 3
**Originality:** 3
**Overall Recommendation:** 4
**Confidence:** 3

**Summary:**

This paper introduces an interesting theory for modeling data acquisition and pricing in large-scale ML training. Specifically, it assumes there are $k$ data sellers, each with a data source $D_i$ and a price $c_i$. The buyer has a fixed budget to purchase data from these sources, with the goal of minimizing the loss of the trained model. The main challenge is that training the model can itself be computationally expensive, yet the buyer needs to decide the purchasing allocation based on how the acquired data would affect the final loss.

To deal with this dilemma, the paper introduces a novel framework for estimating the optimal allocation without actually training the model, using only certain summary information about the data sources. Concretely, the paper focuses exclusively on the realizable linear model with squared loss. In this setting, it shows that one can determine the optimal allocation based only on the covariance matrices of the data sources, without needing access to the true labels. The paper then extends the setup to a mechanism-design framework, where pricing depends on the buyer’s allocation strategy.

**Compliance With Llm Reviewing Policy:**

Affirmed.

**Final Justification:**

I keep my score as is. See Rebuttal Acknowledgement.

**Key Questions For Authors:**

See weaknesses section above.

**Strengths And Weaknesses:**

**Strengths:**

1. In my opinion, the idea of finding the optimal data allocation without training is a genuinely interesting conceptual contribution. It has a similar flavor to computing certain functionals of the data without fully revealing the data themselves.
2. The toy model based on the linear realizable model with squared loss is an interesting starting point and clearly demonstrates the paper’s main conceptual leap.
3. Although I am not an expert in mechanism design, the paper’s contribution on that side seems non-trivial to me.

**Weaknesses:**

1. My main reservation about the paper is that the assumption that one can estimate the gradient of the loss is quite strong. It essentially requires the data sellers to reveal certain private information about their data, such as the covariance matrix in the linear-model case. This is difficult to justify in practice. It is not clear why a seller would be willing to reveal even unlabeled information about the data before the sale.
2. For the framework to be meaningful, the summary statistics needed to find the optimal allocation must contain strictly less information than what is needed to recover the optimal model parameter. Otherwise, the seller would have little reason to sell the data itself. In the linear-model setting, this can be justified because the labels, which are necessary for training but not for optimal allocation, may be the costly part of the data. However, I do not see how this appealing separation extends to more general parametric model classes.
3. The paper’s strongest positive result seems to rely heavily on the realizable linear model with squared loss. As a result, it is unclear to me how much of the framework would survive beyond this highly stylized setting, and whether the main conceptual insight can be carried over to more realistic modern ML models.

---

> ### Author Rebuttal · Authors · 2026-03-31
>
> We thank the reviewer for their thoughtful comments and feedback about our work. We address the feedback individually below, and look forward to incorporating these suggestions into our next revision:
>
> >Realism of model (W1-2): We appreciate these points from the reviewer, and wish to clarify the reasons for which we think that the gradient estimation oracle and the overall setup are in fact stylistically very relevant, even if not exactly what happens in practice. In fact, we view the “gradient estimation oracle” as one of the contributions of our work, so we are eager to make these clarifications in the next revision of our paper as well:
>
> First, the setting we aim to capture is the one where the data buyer can actually see all of the data, but must pay a price to actually use it in a commercial model (e.g., OpenAI paying the NYT for their data, or Anthropic negotiating a deal with Reddit); in these cases, the data is fully available to train on, but due to legal constraints, data pricing is still necessary.
>
> It is indeed convenient that for OLS there is a separation between inputs and labels, but (a) this is not strictly necessary for the model to be realistic (as per #1 above); (b) even in the absence of costly labels, one can imagine using a surrogate model to approximate coarse labels and subsequently compute the gradient, which would correspond to a valid “noisy oracle” as per our model.
>
> Finally, we note that our results capture a variety of realistic data filtering methods (e.g., influence function-based methods, data quality-based methods, etc.), which we can conceptually think of as gradient estimates of varying quality.
>
> >Reliance on OLS (W3). We expect our results to extend asymptotically to linear data-generating processes more generally (as in [KMT'24]), but we agree in general that such model classes may not be reflective of the reality of training large language models, for example. The key issue here is actually that we compare the data selection found by the buyer to the *globally optimal* data selection, which for more complicated model classes, is not generally attained. Indeed, if we instead ask for a locally optimal data selection, then we can leverage results from non-convex optimization to attain similar positive results. However, given that the main conceptual contribution of our paper is the link between data selection and optimization with noisy gradient signal, we felt that this extra complexity was not warranted.
>
> [KMT'24]: Towards a Statistical Theory of Data Selection under Weak Supervision, Kolossov G., Montanari A., Tandon P., ICLR'24

---

> > ### Author Rebuttal · Reviewer_db6A · 2026-04-01
> >
> > I thank the authors for the responses. I now understand that the authors’ motivation is a setting where the data may be publicly accessible, but commercial use is forbidden unless the buyer pays. This clarification helps. However, this setup still looks somewhat weird to me in practice. In particular, it is unclear how the data seller would even know, or prove, that the data was used commercially without payment. For this reason, I still find the practical motivation for this setting not fully convincing.

---

### Official Review · Reviewer_Rjw8 · 2026-03-11

**Soundness:** 4
**Presentation:** 3
**Significance:** 4
**Originality:** 4
**Overall Recommendation:** 5
**Confidence:** 1

**Summary:**

Training large-scale machine learning models requires large amounts of high-quality data. Data providers want fair compensation based on the quality of their contributed data. It makes sense for AI companies to pay data owners according to their contribution to the model, but evaluating such contributions requires repeated retraining, which is computationally infeasible in practice. To address this problem, this work studies data pricing under compute constraints and proposes a framework called “pricing with an attribution oracle”. The proposed method estimates each data source's contribution to the model based on its gradient contribution to the loss function, without requiring any retraining. Theoretical results show that the framework admits an optimal closed-form solution. However, the optimal data selection policy depends heavily on the cost of computation and the heterogeneity of data sources. Moreover, compute-constrained policies can break incentive compatibility. To resolve this issue, the authors introduce a strongly convex regularization term into the objective function, which restores approximate monotonicity and enables the design of truthful mechanisms.

**Compliance With Llm Reviewing Policy:**

Affirmed.

**Final Justification:**

Thanks, the author's response addressed my concern.

**Key Questions For Authors:**

No.

**Limitations:**

Yes

**Strengths And Weaknesses:**

Strengths:
•The introduction of the gradient approximation oracle transforms the expensive data pricing problem, which traditionally requires retraining to evaluate each data source's contribution, into a more computationally efficient framework without lossing fidelity. The fidelity of the approximation is governed by the hyperparameter ε, which can be manually tuned based on computational budget constraints and accuracy requirements.

•The proofs are both intuitive and rigorous, with each step clearly motivated. Furthermore, the tradeoff among competing requirements and metrics is thoroughly characterized in the Compute-Truthfulness-Utility Trilemma, and the discussion provides valuable insights.

Weaknesses:

The authors use the Poisson distribution in the definitions and setup section. While this may simplify the mathematical derivations, is there any literature or theoretical justification supporting this distributional assumption?

---

> ### Author Rebuttal · Authors · 2026-03-31
>
> We would like to thank the reviewer for their positive feedback. As you correctly pointed out, Poisson sampling, is primarily used for the purposes of the analysis. Independent Poisson counts simplify the derivations, and conditional on the total sample count the model becomes multinomial (Lemma 4). Appendix B.4 then relates the Poissonized objective back to the surrogate objective via concentration bounds. We will add a short explanation of this motivation in the main text so the modeling choice is less abrupt.

---

> > ### Author Rebuttal · Reviewer_Rjw8 · 2026-04-03
> >
> > Thank you for your response. My concern is fully adressed.

---

### Official Review · Reviewer_hQRo · 2026-03-12

**Soundness:** 3
**Presentation:** 3
**Significance:** 3
**Originality:** 3
**Overall Recommendation:** 4
**Confidence:** 4

**Summary:**

This paper discusses how to select data from multiple data sources and determine appropriate prices under computation constraints. The authors propose a unified framework that combines data selection, compute constraints, and data pricing, and analyze the optimal strategies for data acquisition in this setting.

**Compliance With Llm Reviewing Policy:**

Affirmed.

**Final Justification:**

In the rebuttal phase, the authors addressed my questions, and I think the quality of this paper is good. Therefore, I enhanced my score. Thanks!

**Key Questions For Authors:**

I also have a question regarding the relationship between the data distributions provided by sellers and the resulting expected loss or downstream performance. As I understand the paper, the underlying logic is that the data distribution affects the statistical structure of the training data, which in turn determines the information matrix and ultimately influences the expected loss. If this interpretation is correct, then the validity of the framework seems to depend on a rather specific modeling assumption. In particular, I wonder whether the proposed approach is inherently limited to certain classes of models or learning settings where such a distribution-to-performance link can be explicitly characterized. This may constrain the generality and practical applicability of the framework beyond the stylized setting considered in the paper.

**Limitations:**

The practical applicability of this work is limited. For example, the framework assumes that the data distributions of different sources are known in advance, which is often unrealistic in real-world scenarios.

**Strengths And Weaknesses:**

Strengths:

- The authors add the budget to the optimization objective and propose data attribution oracle for this new concept.

- The theoretical analysis of this paper is good.

- The authors connect data selection, budget within budget, and mechanism design together and provide a foundation for future research in the data market.

Weaknesses:

- Although I know this paper is a theoretical framework paper, some empirical analysis still needs to verify the efficacy of the approach.

- Technical novelty is another concern. The paper mainly presents an incremental study that combines existing techniques from data selection, compute-aware optimization, and mechanism design, rather than introducing fundamentally new methods or theoretical tools.

- Some parts of the writing in this paper are not good. Rather than directly introducing the settings of this paper, the authors should first introduce the research gaps in comparison with previous studies and emphasize the motivation of this paper.

---

> ### Author Rebuttal · Authors · 2026-03-31
>
> We thank the reviewer for taking the time to read our paper and for providing constructive feedback. We respond to the points they raised below.
>
> >Empirical analysis: The main motivation of our work is to establish a principled framework in which we can make theoretical claims and comparisons between different data pricing and acquisition schemes. That said, we believe that insights from our work can inform practical algorithms. For instance, the result related to backloading the compute budget is something that we expect to hold beyond the setting we study in our work. We will work on experimental results for the next version of our manuscript.
>
> >Technical novelty: We view the unified treatment of the data selection problem (under limited compute budget) with the data acquisition problem from strategic sources as the main novelty of our setting. Our results show that one cannot straightforwardly combine tools from these two lines of work to get results in our setting. Indeed, using approximate data selection policies (due to compute budget constraints) makes the allocation rule non-monotone, which breaks incentive compatibility (as shown in our work). Thus, we extend prior work on mechanism design that works for monotone data selection rules to non-monotone selection rules. Moreover, we develop tools that could find applications outside the scope of our work, like the data attribution oracle in Section 3.2. Overall, a unified treatment of these issues combines tools from numerical linear algebra, optimal design, mechanism design, and optimization.
>
> >Writing of the paper:  We thank the reviewer for the suggestion, and we will revise our draft accordingly. Our main contribution is a theory/framework on data pricing under compute constraints. The core point is that compute is not just an implementation detail: once the buyer cannot solve the acquisition problem exactly, the interaction between optimization, approximation quality, and incentives becomes qualitatively different. In particular, the paper shows that compute-constrained/early-stopped allocation rules can lose monotonicity even when the exact optimum is monotone, which in turn breaks truthful payment design; we then show how to restore approximate monotonicity and compute payments with essentially no extra overhead. Prior work studies non-strategic data selection under compute constraints (mostly from an empirical point of view), or strategic data acquisition without compute constraints. Thus, the main gap our work fills is to bridge these two lines of work from a theoretical point of view.  We will revise the introduction to foreground this research gap more clearly relative to prior work on data selection, compute-aware optimization, and mechanism design.
>
> >Relationship between data distributions and downstream performance: Indeed, in this setting there is a closed-form link between the structure of the data distribution and the downstream performance. Nevertheless, even when this link does not exist in closed form, in practice we know that the composition of the data distribution heavily influences the downstream performance. In both our work and in practice, the interaction of the data buyer with the distribution is only through the approximate gradient oracle—in this way, we still believe our results are useful for providing practical intuition.
>
> >On distributions being known: Thank you for bringing this up. In many applications, the data buyer can indeed get an understanding of the data distribution that the seller is providing. For instance, buyers wish to acquire copyrighted content to train their models on. Hence, the buyers have a very good understanding of how the underlying data look (e.g., their quality, diversity etc.) and they need to pay for the right to use the data. In other applications, data providers might approach data buyers and offer them small samples of their dataset so that they can figure out how much they can help improve their model. This mirrors the classical line of work on mechanism design where the main challenge is not the lack of information from the buyer’s side about the value of the goods they are trying to purchase, but rather the strategic nature of the seller. That said, we agree that it is an interesting direction to extend our ideas to settings with unknown distributions. We will elaborate on this discussion in the next version of our work.

---

> > ### Author Rebuttal · Reviewer_hQRo · 2026-04-03
> >
> > My questions are addressed. Thanks!

---

### Decision · Program_Chairs · 2026-04-30

**Decision:**

Accept (regular)

**Comment:**

This paper studies data acquisition and pricing under compute constraints, with a mechanism-design perspective on truthful procurement. Reviewers were overall positive and agreed that this is a timely and meaningful problem. Reviewers appreciated the formulation, found the gradient-oracle perspective conceptually interesting, and viewed the compute-truthfulness-utility tradeoff as a useful theoretical lens. Several reviews also noted that the paper provides a substantial and technically meaningful theoretical analysis.

When it comes to limitations, reviewers noted that the framework assumes relatively strong access to seller distributions or informative summaries, and that it remains unclear how directly the model maps onto modern large-scale training pipelines. Some reviewers also wanted stronger empirical support. These are important caveats and should be stated clearly in the final version. Still, the contribution appears to be a useful theoretical foundation, and the balance of the discussion favors acceptance. I therefore recommend acceptance.